# Harmonic-Percussive Disentangled Neural Audio Codec for Bandwidth Extension

## Abstract

Bandwidth extension, the task of reconstructing the high-frequency components of an audio signal from its low-pass counterpart, is a long-standing problem in audio processing. While traditional approaches have evolved alongside the broader trends in signal processing, recent advances in neural architectures have significantly improved performance across a wide range of audio tasks. In this work, we extend these advances by framing bandwidth extension as an audio token prediction problem. Specifically, we train a transformer-based language model on the discrete representations produced by a disentangled neural audio codec, where the disentanglement is guided by a Harmonic–Percussive decomposition of the input signals, highlighting spectral structures particularly relevant for bandwidth extension. Our approach introduces a novel codec design that explicitly accounts for the downstream token prediction task, enabling a more effective coupling between codec structure and transformer modeling. This joint design yields high-quality reconstructions of the original signal, as measured by both objective metrics and subjective evaluations. These results highlight the importance of aligning codec disentanglement and representation learning with the generative modeling stage, and demonstrate the potential of global, representation-aware design for advancing bandwidth extension.

## 1 Introduction

Bandwidth extension seeks to reconstruct the high-frequency content of a signal from its low-frequency representation. This problem can be viewed as a form of inpainting where missing information is recovered from degraded observations, and later extended to audio through spectrogram-based methods. Applications arise in telecommunication, where it improves speech quality (Chennoukh et al., 2001), as well as in music restoration (Moliner & Välimäki, 2022). While early methods relied on handcrafted signal processing techniques (Dietz et al., 2002), recent approaches leverage neural networks (Pulakka & Alku, 2011) that can learn efficient internal representations and achieve significantly better results.

On a more general standpoint, representation learning for audio has benefited from encoder–decoder architectures (Oja, 1982), particularly neural codecs based on the VQ-VAE paradigm with quantization (Van Den Oord et al., 2017). These models, originally motivated by compression, yield low-bitrate latent representations while preserving high reconstruction quality. Beyond compression, the resulting latents have proven effective for downstream tasks (Borsos et al., 2023). In parallel, advances in transformer architectures and large language models, combined with neural audio codecs, have established strong baselines in both speech (Wang et al., 2023a) and music generation (Copet et al., 2023).

Despite their effectiveness, codec-based representations often lack interpretability and versatility. This has motivated refined architectures that enforce semantically meaningful latents by reshaping training objectives and model design. Such disentangled representations, tailored to downstream tasks, can be exploited to improve task performance. Disentanglement has been explored in diverse audio processing settings, from specialized applications (Takahashi et al., 2021) to more general approaches (Hsu et al., 2023).

In this work, we leverage recent advances in language modeling over neural audio codec representations to address the bandwidth extension task. Our goal is to redefine codec design by shaping

the latent space for downstream applications and enhancing interpretability. While prior studies have often treated codec structure as fixed and external to language model training, we propose to integrate codec design directly into the prediction pipeline.

We first design the Harmonic-Percussive disentangled codec (HP-codec), a neural codec that explicitly separates high- and low-frequency components and further factors the latent space to capture harmonic and percussive structures, whose characteristic cross-band patterns improve the predictability of high-frequency components from their low-frequency counterparts. Building on this representation, we introduce HP-codecX, a bandwidth extension model based on a transformer language model whose architecture is adapted to HP-codec's structure. The model is trained to predict high-frequency content from HP-codec's low-frequency latents, thereby addressing the bandwidth extension task.

Our main contributions are as follows: **(1)** We introduce HP-codec, a semantically informed disentangled neural audio codec that leverages an Harmonic–Percussive decomposition of audio signals. **(2)** We adapt its latent representation to a language modeling task aligned with bandwidth extension. **(3)** We design and train a multi-branch language model tailored for bandwidth extension. **(4)** We demonstrate state-of-the-art performance on bandwidth extension, with consistent improvements in both objective metrics and human listening tests.

## 2 RELATED WORK

### 2.1 NEURAL AUDIO CODECS AND DISENTANGLEMENT

Feature extraction from audio has long been studied, beginning with handcrafted mathematical representations such as the Fourier transform, and later perceptually motivated features like the Mel scale (Stevens et al., 1937). With the advent of neural networks, representation learning shifted toward autoencoders (Kingma & Welling, 2014), followed by the introduction of residual vector quantization (RVQ) between encoder and decoder (Van Den Oord et al., 2017). Modern neural audio codecs combine an encoder, RVQ, and decoder, trained with composite objectives often including adversarial losses (Zeghidour et al., 2021; Défossez et al., 2023; Kumar et al., 2023). These models currently define the state of the art in audio compression, achieving high reconstruction quality at low bitrates.

Subsequent research has refined codec architectures to address specific limitations. For example, Takida et al. (2022) proposed a differentiable quantization mechanism to eliminate the stop-gradient trick. Yang et al. (2023) introduced group residual quantization, reducing the number of quantizers required for high-quality reconstruction. Liu et al. (2024b) separated encoding into semantic and acoustic components, enabling operation at very low bitrates and facilitating language model integration. The Mimi codec (Défossez et al., 2024) augmented RVQ with a parallel quantizer to distill semantic information, improving phonetic discriminability.

Beyond achieving high compression rates, neural audio codec representations have proven valuable for downstream tasks. The utility of discrete latents was first demonstrated in computer vision, where convolutional models trained on VQ-VAE representations enabled high-quality image generation (Razavi et al., 2019). Extending this principle to audio has motivated task-specific codec designs that enforce disentangled and semantically meaningful representations. For example, Takahashi et al. (2021) designed a codec for singing voice conversion that separates pitch, amplitude, and singer identity from acoustic information, Wang et al. (2023b) disentangled speaker identity and timbre for zero-shot adaptive speech generation, and Polyak et al. (2021) separated prosody, speaker identity, and pitch for speech resynthesis. Other works impose disentanglement through auxiliary objectives, such as Omran et al. (2023) for speech separation or Ju et al. (2025), which constrains a multi-branch quantizer with pretext tasks and gradient reverse tricks for zero-shot speech synthesis. More general approaches aim to build codecs that support multiple data modalities and subtasks, by separating speech, music, and environmental sounds (Bie et al., 2025; Jiang et al., 2025), or disentangling frequency bands (Luo et al., 2024; Giniès et al., 2025).

## 2.2 LANGUAGE MODELS FOR AUDIO APPLICATION

The success of the Transformer architecture (Vaswani et al., 2017) and its subsequent adoption in language models for next-token prediction (Devlin et al., 2019) has motivated a growing line of work treating discrete audio representations as tokens for language modeling. Baevski et al. (2020) demonstrated this approach by combining a Transformer-based language model with masked encoder latents and contrastive learning, yielding robust discrete audio representations. Building on this idea, Huang et al. (2022) integrated masking strategies with Transformer blocks and neural audio codecs to construct latent representations well-suited for classification tasks. Beyond representation learning, language models have also been shown to be effective for generative audio modeling. For instance, Wang et al. (2023a) leveraged codec-derived discrete units with language models for zero-shot text-to-speech, an idea later extended to speech translation (Zhang et al., 2023). Similarly, Copet et al. (2023) applied next-token prediction to music generation, further underscoring the generality of this paradigm. A related strategy has recently been explored in speech restoration, where generative language models are trained to predict clean codec tokens from their degraded versions (Li et al., 2024; Yang et al., 2024).

## 2.3 BANDWIDTH EXTENSION

Bandwidth extension, which consists in inferring high-frequency content from low-pass signals, has been studied extensively. Classical approaches focused on spectral manipulation, such as duplicating or rescaling low-frequency spectra into higher bands (Dietz et al., 2002; Nagel & Disch, 2009). Neural methods substantially reshaped the problem, with early applications of U-Nets for reconstructing truncated signals (Kuleshov et al., 2017). Diffusion-based approaches further advanced performance, including NU-Wave (Lee & Han, 2021; Han & Lee, 2022) and AudioSR (Liu et al., 2024a), which reconstruct high-frequency details from waveform or mel-spectrogram inputs. Related tasks such as image inpainting have also been addressed with autoregressive models applied to VQ-VAE representations (Peng et al., 2021). Diffusion processes have also been coupled with neural audio codecs, using a MAMBA-based (Gu & Dao, 2024) token enrichment for speech enhancement (Fang et al., 2025). Some hybrid methods combine differentiable digital signal processing (Engel et al., 2020) with neural networks (Grumiaux & Lagrange, 2023). Li & Luo (2025) base their architecture on the codec of (Luo et al., 2024), replacing a processing step between the encoder and the decoder by a transformer model to predict missing information.

## 3 OUR APPROACH

We denote by $s$ a time-domain signal, and by $s_{b;SR}$ its version band-limited to $b$ kHz and sampled at the sampling frequency $SR$. With this notation, $s_{8;16}$ corresponds to the signal $s$, band-limited to 8 kHz and sampled at 16 kHz. The goal of bandwidth extension is to reconstruct $s_{24;48}$ (the same signal with frequency content up to 24 kHz and sampled at 48 kHz), from the low-frequency components in $s_{8;16}$.

To this end, we leverage the generative modeling capabilities of transformer-based language models by operating in a discrete token space. Neural audio codecs provide compact discrete representations that are well suited for such models, enabling the use of NLP-style sequence modeling techniques. While discretization necessarily discards some fine-grained information, it also removes low-level variability that can complicate learning, reduce generalization, or induce artifacts. In practice, the codec representation yields a cleaner and more tractable modeling domain in which the transformer can focus on predicting the missing high-frequency structure.

We thus propose a two-stage neural architecture that combines a disentangled neural audio codec (HP-codec) with a language model to form our bandwidth extension model (HP-codecX). HP-codec is first trained to produce a structured latent representation of the input signal, after which the language model is fitted on this latent space to capture and predict the missing high-frequency information. The overall framework is illustrated in Fig. 1 and Fig. 2. Audio examples are given at https://harmonic-percussive-bandwidth-extension.github.io/.

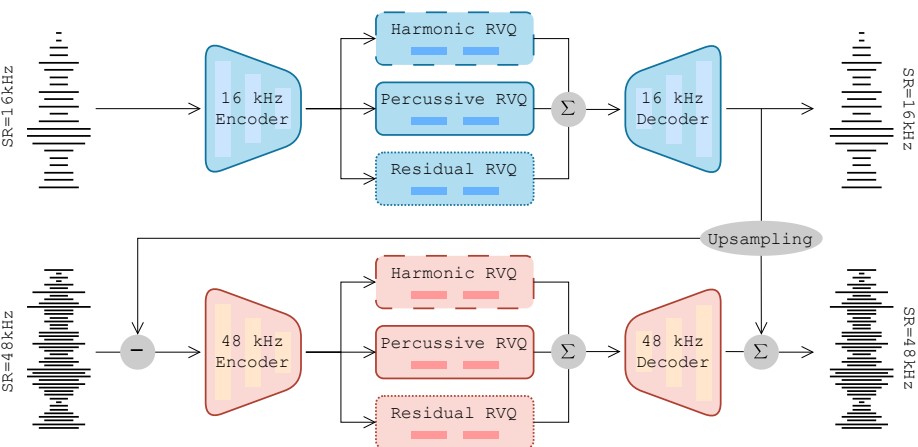

Figure 1: **HP-codec**, our spectrally informed disentangled codec. It is divided in two branches operating at different sampling rates: a 16 kHz branch and a 48 kHz branch. Each branch contains parallel RVQs which are composed of a harmonic section, a percussive section and a residual section.

### 3.1 HP-codec: the Disentangled Codec

#### 3.1.1 Frequency disentanglement

Our codec builds upon the architecture of Giniès et al. (2025), itself derived from a low-bitrate variant of the DAC codec (Kumar et al., 2023), by introducing a branched design. Specifically, the DAC structure is replicated into two branches: one dedicated to encoding and reconstructing low-frequency components, and the other to high-frequency components. This design enforces a disentanglement of frequency bands in the learned discrete representations, while maintaining a dependency between them. The dependency between frequency bands is enforced by computing the residual between the output of the low-frequency branch and the input to the high-frequency branch, as illustrated in Fig. 1.

In our implementation, the first branch operates at a 16 kHz sampling rate, modeling spectral components up to 8 kHz, while the second branch operates at 48 kHz to capture the remaining content up to 24 kHz. Denoting by $\hat{s}_{8;16}$ the reconstruction extracted from the first branch and by $\hat{s}_{8;48}$ its upsampling to 48 kHz, the input to the second branch is defined as the residual $s_{24;48} - \hat{s}_{8;48}$. To ensure compatibility between branches, the compression ratios of the 16 kHz and 48 kHz branches are selected such that both produce the same number of tokens per signal (i.e. each token corresponds to the same temporal context across branches). In our setting, each RVQ contains two consecutive codebooks.

#### 3.1.2 Semantically informed sections

To strengthen the spectral structure shared across the two branches of HP-codec, we further decompose the RVQs into three parallel modules: a harmonic RVQ, a percussive RVQ, and a residual RVQ, as shown in Fig. 1. Each module is specialized for encoding harmonic, percussive, and residual components of the signal, respectively. This design is motivated by the relevance of Harmonic + Noise decompositions for modeling speech and audio signals (Serra & Smith, 1990; McAulay & Quatieri, 1992; Richard & d'Alessandro, 1996; Fitzgerald, 2010; Driedger et al., 2014) and follows the line of work adapting neural architectures to the specificities of audio signals (Pons et al., 2016). Beyond improving the interpretability of the learned latent space, this decomposition reinforces the coupling between the low- and high-frequency branches: harmonic structures in the low-frequency band are closely correlated with their high-frequency counterparts, and the same holds for percussive components.

After quantization, the discrete representations produced by the three sections of each branch are summed and subsequently passed to the decoder, which synthesizes the corresponding time-domain reconstruction.

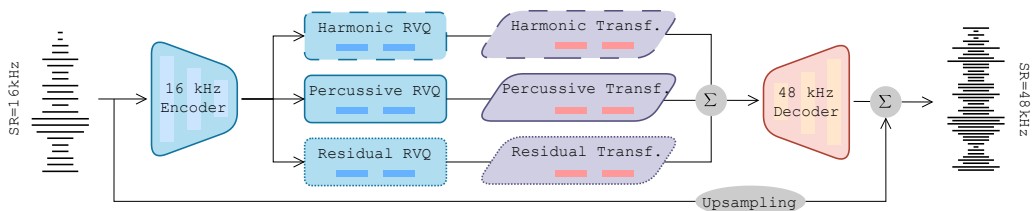

Figure 2: **HP-codecX**, our bandwidth extension model. It connects the 16 kHz representation, extracted from the input, to the 48 kHz decoder, through a language model organized into three sub-models: a harmonic estimator, a percussive estimator and a residual estimator.

### 3.1.3 TRAINING PROCEDURE

HP-codec is optimized using a combination of multiscale Mel-spectrogram losses (to preserve spectral fidelity), codebook and commitment losses (to regularize the RVQs and align them with encoder outputs), as well as feature matching and adversarial losses computed with multi-period and multi-scale STFT discriminators (Kumar et al., 2023). Training follows the cascade strategy of Giniès et al. (2025): we first train the low-frequency branch, then freeze its parameters, and train the high-frequency branch. Finally, we jointly finetune the entire codec. Loss scaling across training phases is also left untouched: DAC's scheme (Kumar et al., 2023) is applied in the first two phases; during finetuning, we aggregate codebook-related losses by summation across branches, while averaging the remaining losses. All phases use an exponential learning-rate scheduler with decay factor $\gamma = 0.999996$, with a base rate of $10^{-4}$ in regular phases and $5 \times 10^{-5}$ in finetuning.

At each training step, we uniformly sample from {harmonic, percussive, full} to determine the training objective. In a full iteration, all RVQ sections are updated using batches of non-decomposed signals. In a harmonic iteration, only the harmonic section of each RVQ is trained, with inputs derived from harmonic–percussive–residual decomposition (Driedger et al., 2014) restricted to the harmonic components. The same procedure is applied for percussive iterations. The residual sections are updated exclusively during full iterations, ensuring that they capture signal structures not explained by the harmonic or percussive sections. For each branch, the overall codebook and commitment losses are computed as the sum of the corresponding losses across its RVQ sections.

### 3.2 HP-CODECX: LANGUAGE MODEL FOR PREDICTION

#### 3.2.1 LANGUAGE MODEL

In the bandwidth extension setting, we only observe $s_{8;16}$, from which we extract the Harmonic, Percussive, and Residual token sequences from the 16 kHz branch: $\{H_n^{16;1}, H_n^{16;2}\}$ for the first and second Harmonic codebooks, $\{P_n^{16;1}, P_n^{16;2}\}$ for the Percussive codebooks, and $\{R_n^{16;1}, R_n^{16;2}\}$ for the Residual codebooks, with $n \in \{1, \dots, N\}$ indexing the tokens within each sequence.

Following Wang et al. (2023a), we adapt an autoregressive transformer decoder to operate on the tokens of each RVQ section (Fig. 2). For instance, the harmonic transformer takes as input $\{(H_n^{16;1}), (H_n^{16;2})\}$ and predicts $(\tilde{H}_n^{48;1})$ an estimate of the tokens of the high frequency branch first codebook. In a second stage, the model uses $\{(H_n^{16;1}), (H_n^{16;2}), (\tilde{H}_n^{48;1})\}$ to predict $(\tilde{H}_n^{48;2})$. This procedure is applied analogously to the percussive and residual sections.

The prediction task is decomposed into three subtasks, yielding the estimated token sequences $\{(\tilde{H}_n^{48;1}), (\tilde{H}_n^{48;2})\}$, $\{(\tilde{P}_n^{48;1}), (\tilde{P}_n^{48;2})\}$, and $\{(\tilde{R}_n^{48;1}), (\tilde{R}_n^{48;2})\}$. These are summed and passed through the 48 kHz decoder to reconstruct the high-frequency components, which are then added to $s_{8;48}$ to produce $\tilde{s}_{24;48}$, an estimate of the full-band signal $s_{24;48}$.

#### 3.2.2 TRAINING PROCEDURE

All transformer modules of HP-codecX, corresponding to the RVQ sections, are trained using a standard cross-entropy objective and optimized jointly, with the total loss defined as the sum of the cross-entropy terms from each prediction. Following Wang et al. (2023a), we train the two-stage

prediction process by uniformly sampling from $\{1, 2\}$ at each iteration to determine which stage is updated. The training uses a cosine annealing learning rate schedule with an initial rate of $10^{-4}$.

# 4 EXPERIMENTAL SETUP AND RESULTS

## 4.1 BASELINES

As comparison references for the performances of HP-codecX, we chose to compare to the Apollo model (Li & Luo, 2025), which reshapes the GULL model (Luo et al., 2024) for bandwidth extension, and to the AudioSR model (Liu et al., 2024a), which performs bandwidth extension through a diffusion process applied to the spectrogram of the signals. Apart from slight differences in the approaches of these models and ours (the Apollo model works at 44.1 kHz and is trained on degraded audios encoded through MP3 encoders at low bitrates and the AudioSR model is working at 48 kHz and is trained on signals passed through various low pass filters), we estimated that training conditions were sufficiently similar to allow for a proper comparison between all models.

## 4.2 DATASETS

The training of our model has been performed on the training part of the MUSDB18 dataset (Stöter et al., 2018) and on the JAMENDO dataset (Bogdanov et al., 2019). The testing of HP-codec was performed on the testing part of MUSDB18 dataset (Stöter et al., 2018). The testing of our bandwidth extension model's (HP-codecX) prediction was done on the testing part of the MUSDB18 dataset (Stöter et al., 2018), as well as on datasets that were not observed during training: the ENST-Drums dataset (Gillet & Richard, 2006), the OrchideaSOL dataset (Cella et al., 2020), the Medley-solos-DB dataset (Lostanlen et al., 2018), and on a Monophonic synthetic dataset and a Polyphonic synthetic dataset which were built according to the implementation designed in Grumiaux & Lagrange (2023).

The JAMENDO and MUSDB18 datasets are music datasets gathering more than 55,000 music signals in the training set. Our training set gathers almost 3 800 hours of music samples, and our testing set is composed of 1,000 samples randomly extracted from the 50 music tracks from the 3.5 hour long MUSDB18 testing set. The harmonic–percussive–residual decomposition used during training follows the procedure of Driedger et al. (2014). The method applies horizontal and vertical median filtering to the magnitude spectrogram, yielding estimates of the harmonic and percussive components, respectively. The residual component is then defined as the part of the signal not captured by either of these two estimates.

We also constituted testing sets, each composed of 1000 samples extracted from OrchideaSOL and Medley-solos-DB datasets (which gather single instruments recordings), from ENST-Drums dataset (which gathers drums recordings) and from the Monophonic and Polyphonic synthetic datasets (which are composed respectively of purely harmonic sources and superposition of many harmonic sources). These testing sets were used for out-of-domain testing.

All samples are recorded at 44.1 kHz, upsampled at 48 kHz and contain information up to 22.05 kHz.

## 4.3 OBJECTIVE METRICS

We evaluate reconstruction quality using a combination of spectral, waveform, and perceptual metrics. Specifically, we adopt the multiresolution Mel- and STFT-losses from Kumar et al. (2023) to capture spectral discrepancies, an $\ell_1$ waveform loss to assess sample-level fidelity, and the ViSQOL metric (Chinen et al., 2020) as a proxy for perceptual quality. We additionally report the scale-invariant signal-to-distortion ratio (SI-SDR) (Le Roux et al., 2019) as a measure of distortion relative to the underlying content. While commonly used in audio coding, it is less suited for synthesis tasks, as its sensitivity may penalize samples that remain perceptually acceptable as stated in Défossez et al. (2024) and Parker et al. (2025).

## 4.4 LISTENING TEST

We evaluated the perceptual quality of our bandwidth extension approach using a MUSHRA test (Schoeffler et al., 2018). The study involved 15 non-expert participants under standard office con-

ditions, using headphones and with the option to replay excerpts. Each participant rated 12 sets of 5 signals on a 0–100 scale with respect to a reference. For each input $s$, the test set included the anchor $s_{8,16}$, the reference $s_{24,48}$, and three system outputs: $\tilde{s}_{24,48}^{Apo}$, $\tilde{s}_{24,48}^{Aud}$, and $\tilde{s}_{24,48}^{HPX}$. The 12 excerpts were randomly sampled from the MUSDB18 test set, with half drawn from segments exhibiting high energy in the high-frequency band and half from the remaining samples.

### 4.5 TECHNICAL SPECIFICATIONS

HP-codec follows the DAC architecture (Kumar et al., 2023), with modifications to the number of tokens and encoder/decoder rates. For the 16 kHz branch, we use encoder rates of $\{2, 2, 5, 8\}$ with two codebooks per RVQ, resulting in a bitrate of 6 kbit/s and a compression ratio of 42.6. For the 48 kHz branch, we adopt encoder rates of $\{2, 5, 6, 8\}$ to preserve proportionality with the sampling rates, yielding a bitrate of 12 kbit/s and a compression ratio of 64. These settings deliberately operate in a low-bitrate regime, reflecting a tradeoff between codec reconstruction quality and the predictive capacity of the language model. We train on 0.38-second audio sequences with a batch size of 32 for the first branch and 16 for the second branch, as well as during finetuning. The model is trained for 26 hours on a single NVIDIA L40S GPU with 48 GB of memory.

The transformer modules of HP-codecX follow the autoregressive design of Wang et al. (2023a). Each module employs three input embeddings mapping HP-codec tokens to 1024-dimensional representations, a 6-layer transformer decoder with 8 attention heads and hidden dimension 4096, followed by two dense layers that output the following tokens' prediction. Training is performed on 2.5-second audio samples with batches of 32, on a single NVIDIA L40S 48 GB GPU for 54 hours.

### 4.6 TESTING HP-CODEC

To verify that introducing semantic sections does not degrade the reconstruction quality of HP-codec, we evaluate the model on the MUSDB18 (Stöter et al., 2018) test set. For comparison, we adapt the disentangled codec of Giniès et al. (2025) to operate at 48 kHz under identical compression rates, and we retrain a DAC model (Kumar et al., 2023) on 48 kHz audio at the same compression ratio. Since DAC is a widely used baseline with extensive comparisons in the literature, including it provides a clearer sense of how HP-codec aligns with existing methods. The results are summarized in Table 1.

Table 1: Reconstruction metrics ($\pm$ standard deviation) for HP-codec. The reference model is a modified version of Giniès et al. (2025) in which the harmonic, percussive, and residual components are removed. DAC-48kHz denotes a DAC model Kumar et al. (2023) retrained on our dataset. Both comparison models operate at 48 kHz and use the same compression rate as our model.

| Sampling rates | HP-codec | | Reference | | DAC-48kHz |
|---|---|---|---|---|---|
| | 16000 | 48000 | 16000 | 48000 | 48000 |
| Mel $\downarrow$ | 0.80±0.08 | 0.79±0.05 | 0.70±0.08 | 0.72±0.06 | 0.75±0.08 |
| STFT $\downarrow$ | 2.30±0.29 | 2.29±0.29 | 2.11±0.27 | 2.22±0.28 | 2.24±0.28 |
| Waveform $\downarrow$ | 0.051±0.015 | 0.052±0.015 | 0.041±0.014 | 0.043±0.014 | 0.041±0.013 |
| SI-SDR $\uparrow$ | 6.74±2.53 | 6.30±2.51 | 8.75±2.94 | 8.10±2.91 | 8.79±2.92 |
| ViSQOL $\uparrow$ | 4.33±0.09 | 4.33±0.14 | 4.43±0.07 | 4.33±0.17 | 3.92±0.2 |

These results demonstrate that modifying the RVQ structure to produce a more spectrally informed discrete representation in HP-codec yields performance competitive with both the unmodified reference model and the retrained DAC baseline. In Appendix A, we further show that the semantic sections enhance the interpretability of the learned representations: harmonic sections specialize in reconstructing harmonic content, while percussive sections are better suited for percussive signals.

### 4.7 EVALUATING HP-CODECX

We assessed the quality of the estimated signals using the reconstruction metrics introduced previously, comparing HP-codecX against Apollo and AudioSR. Table 2 reports results for both full-band evaluation (entire signal) and high-frequency evaluation restricted to the $[8\,\mathrm{kHz}, 24\,\mathrm{kHz}]$ range. The

latter is particularly relevant, as Apollo fully reconstructs low-frequency components. Table 3 gathers the results of the perceptual test introduced in Section 4.4. Spectrograms of estimated signals are displayed in Appendix D.

Table 2: Objective reconstruction metrics ($\pm$ standard deviation) for the Apollo (44.1 kHz), AudioSR (48 kHz) models and HP-codecX (48 kHz). The top metrics are calculated over the whole signals (**Global**). The lower metrics calculated on the $[8\ kHz, 24\ kHz]$ band (**HF**).

| **Global** | Apollo (Li & Luo, 2025) | AudioSR (Liu et al., 2024a) | HP-codecX |
|---|---|---|---|
| Mel ↓ | $1.02 \pm 0.12$ | $1.83 \pm 0.43$ | $\mathbf{0.27} \pm 0.09$ |
| STFT ↓ | $3.35 \pm 0.53$ | $3.98 \pm 0.71$ | $\mathbf{1.25} \pm 0.22$ |
| Waveform ↓ | $0.048 \pm 0.013$ | $0.069 \pm 0.019$ | $\mathbf{0.012} \pm 0.007$ |
| SI-SDR ↑ | $3.26 \pm 3.82$ | $13.92 \pm 5.06$ | $\mathbf{19.85} \pm 7.43$ |
| ViSQOL ↑ | $3.26 \pm 0.40$ | $2.98 \pm 0.37$ | $\mathbf{3.58} \pm 0.35$ |

| **HF** | Apollo (Li & Luo, 2025) | AudioSR (Liu et al., 2024a) | HP-codecX |
|---|---|---|---|
| Mel ↓ | $0.80 \pm 0.12$ | $0.83 \pm 0.16$ | $\mathbf{0.49} \pm 0.14$ |
| STFT ↓ | $3.04 \pm 0.51$ | $3.09 \pm 0.54$ | $\mathbf{2.06} \pm 0.40$ |
| Waveform ↓ | $\mathbf{0.008} \pm 0.005$ | $0.010 \pm 0.005$ | $0.012 \pm 0.007$ |
| SI-SDR ↑ | $\mathbf{-28.39} \pm 8.42$ | $-44.20 \pm 10.18$ | $-36.82 \pm 8.40$ |
| ViSQOL ↑ | $3.38 \pm 0.37$ | $3.18 \pm 0.35$ | $\mathbf{3.53} \pm 0.46$ |

HP-codecX consistently outperforms both baselines in reconstructing high-frequency spectral content, highlighting its advantage in capturing fine spectral details. We attribute the low SI-SDR scores to the limitations of the metric in synthesis settings, as this degradation was not reflected in the listening tests.

An evaluation was conducted on the additional testing datasets, using the two baselines and our model, giving the results plotted in Fig. 3. This experiment evaluates the bandwidth extension quality of the three models on previously unseen data types, using out-of-domain datasets.

Table 3: Results of the perceptual evaluation ($\pm$ standard deviation). The MUSHRA test compared Apollo and AudioSR models to HP-codecX. Reference signals ($SR = 48\ kHz$) and anchor signals ($SR = 16\ kHz$) were also evaluated.

| Processing | Score $\pm$ std. |
|---|---|
| Reference | $95.2 \pm 11.1$ |
| HP-codecX | $65.6 \pm 23.2$ |
| AudioSR (Liu et al., 2024a) | $58.7 \pm 23.3$ |
| Apollo (Li & Luo, 2025) | $56.4 \pm 23.2$ |
| Anchor (16 kHz) | $44.8 \pm 23.1$ |

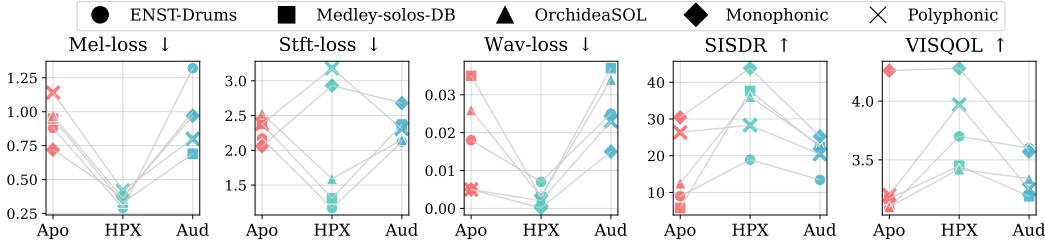

Figure 3: Out-of-domain objective reconstruction metrics. These metrics were computed for the Apollo (Apo), AudioSR (Aud) models and HP-codecX (HPX). They have been calculated at 44.1 kHz for the Apollo model, and 48 kHz for the others.

These results confirm that HP-codecX outperforms both baselines in nearly all cases. In Appendix B, we provide a more detailed evaluation on out-of-domain datasets, further highlighting the strength of HP-codecX in reconstructing percussive and general music signals.

## 4.8 VALIDATING THE SEMANTIC SECTIONS

Tables 4 and 5 highlight the role of the RVQs semantic sections. Using this decomposition of the data together with RVQs was initially motivated by a series of experimental results. Basing the

prediction on the architecture proposed in Giniès et al. (2025), we trained multiple transformer models to predict high-frequency tokens from their low-frequency counterparts. We conducted nine training runs, varying the model depth from 1 to 15 layers and the input duration from 0.33 s to 5 s. Among all experiments, only shallow models exhibited successful learning, with the single-layer transformer achieving the most reliable training behavior. These results suggest that the task complexity induces weak gradient signals which, when propagated through deeper architectures, lead to systematic collapse.

To further validate the choice of semantic decomposition and the split-transformer design, we trained two additional models that omit semantic partitioning of the training data: one using three deep transformers identical to those in HP-codecX, and another using a single transformer shared across the three RVQ branches. As shown in Table 4, both variants and the single-layer reference introduced earlier perform markedly worse than HP-codecX, confirming the advantage of incorporating explicit semantic sections.

Table 4: Objective metrics ($\pm$ standard deviation) for bandwidth-extension with **HP-codecX**, a model with three transformers of equivalent depth trained without semantic decomposition (**EXP1**), a model with a shared transformer trained without semantic decomposition (**EXP2**), and a model without semantic RVQ sections and a single-layer transformer (**EXP3**). Metrics are evaluated on the high-frequency (HF) band.

| **HF** | HP-codecX | EXP1 | EXP2 | EXP3 |
|---|---|---|---|---|
| Mel $\downarrow$ | **0.49**$\pm$0.14 | 0.78$\pm$0.19 | 0.78$\pm$0.18 | 0.62$\pm$0.14 |
| STFT $\downarrow$ | **2.06**$\pm$0.40 | 2.74$\pm$0.60 | 2.84$\pm$0.56 | 2.29$\pm$0.43 |
| Waveform $\downarrow$ | **0.012**$\pm$0.007 | 0.016$\pm$0.012 | 0.017$\pm$0.012 | 0.014$\pm$0.009 |
| SI-SDR $\uparrow$ | -36.82$\pm$8.40 | **-34.92**$\pm$11.74 | -37.89$\pm$10.36 | -37.30$\pm$9.15 |
| ViSQOL $\uparrow$ | **3.53**$\pm$0.46 | 2.60$\pm$0.58 | 2.84$\pm$0.55 | 3.03$\pm$0.45 |

In the experiment illustrated by Table 5, we separately used $\{(\tilde{H}_n^{48;1}), (\tilde{H}_n^{48;2})\}$ the estimated harmonic tokens, $\{(\tilde{P}_n^{48;1}), (\tilde{P}_n^{48;2})\}$ the estimated percussive tokens and $\{(\tilde{R}_n^{48;1}), (\tilde{R}_n^{48;2})\}$ the estimated residual tokens, to evaluate the reconstruction associated with each group of token, and each combination of these groups. We applied this differentiated procedure of estimation to a percussive dataset (ENST-drums), two purely harmonic datasets (Monophonic and Polyphonic) and two general music datasets (Medley-solos-DB and OrchideaSOL). The reconstruction metrics we obtained show that the harmonic section of the estimated tokens better reconstruct a harmonic signal, while the percussive section of the estimated tokens better reconstruct a percussive signal. This strongly underlines the interest of our architecture. Additionally, comparing reconstructions obtained from different token combinations highlights how each section contributes to overall fidelity. For example, harmonic signals are best reconstructed using harmonic tokens alone.

## 5 LIMITATIONS

Our approach is based on two models: HP-codec, a neural audio codec and a language model (which together form HP-codecX). This constitutes the main drawback of this work, as both models must be trained jointly in order to have a fully working process. An alternative line of research would be to avoid architectural coupling at the codec level and instead draw inspiration from recent approaches such as Li et al. (2024) and Yang et al. (2024). These works employ an off-the-shelf codec to produce incomplete discrete token sequences, and then train a generative language model to recover the corresponding clean representations. Although these techniques have so far been explored only in speech domains, they offer a promising direction for future research on the task considered here.

In contrast to various bandwidth-extension systems such as Li & Luo (2025) and Liu et al. (2024a), our model does not support variable input sampling rates and is currently limited to mapping 16 kHz inputs to 48 kHz outputs. This constraint arises primarily from our reliance on discrete audio codecs, which themselves typically operate at fixed sampling rates. Nevertheless, the 16 kHz-48 kHz setting already yields a substantial and practically meaningful improvement in spectral coverage, demonstrating the viability of the proposed framework. Moreover, the relatively low training cost of each model instance makes it feasible to train separate variants for additional sampling rates when needed.

Table 5: Objective metrics (± standard deviation) across datasets, evaluated on high-frequency bands.

| | ENST-drums | Medley-solos-DB | OrchideaSOL | Monophonic | Polyphonic |
|---|---|---|---|---|---|
| **Mel ↓** | | | | | |
| H | 0.41±0.16 | 0.51±0.20 | 0.53±0.35 | 0.33±0.25 | 0.44±0.14 |
| P | 0.40±0.15 | 0.48±0.19 | 0.54±0.35 | 0.37±0.28 | 0.52±0.18 |
| R | 0.89±0.29 | 0.90±0.32 | 0.83±0.48 | 1.67±0.22 | 1.23±0.25 |
| H+P | 0.41±0.16 | 0.50±0.20 | 0.55±0.35 | 0.36±0.24 | 0.49±0.14 |
| H+R | 0.38±0.14 | 0.47±0.17 | 0.47±0.28 | 0.40±0.25 | 0.47±0.12 |
| P+R | 0.39±0.13 | 0.46±0.15 | 0.49±0.28 | 0.46±0.27 | 0.53±0.15 |
| H+P+R | 0.37±0.15 | 0.46±0.18 | 0.50±0.33 | 0.35±0.25 | 0.46±0.14 |
| **STFT ↓** | | | | | |
| H | 1.80±0.47 | 1.90±0.51 | 2.17±0.80 | 2.68±0.54 | 2.89±0.22 |
| P | 1.79±0.49 | 1.86±0.50 | 2.20±0.88 | 2.80±0.65 | 3.17±0.29 |
| R | 2.65±0.63 | 2.59±0.63 | 2.65±1.44 | 5.88±0.99 | 4.96±0.93 |
| H+P | 1.78±0.47 | 1.85±0.51 | 2.18±0.86 | 2.81±0.64 | 3.07±0.28 |
| H+R | 1.71±0.42 | 1.81±0.41 | 1.97±0.63 | 2.93±0.56 | 3.09±0.23 |
| P+R | 1.70±0.44 | 1.80±0.40 | 2.00±0.72 | 3.09±0.64 | 3.33±0.27 |
| H+P+R | 1.69±0.45 | 1.79±0.45 | 2.04±0.78 | 2.78±0.60 | 3.06±0.24 |
| **Waveform ↓** | | | | | |
| H | 0.007±0.007 | 0.002±0.003 | 0.003±0.007 | 0.000±0.001 | 0.002±0.001 |
| P | 0.007±0.007 | 0.002±0.003 | 0.004±0.009 | 0.001±0.003 | 0.003±0.002 |
| R | 0.007±0.005 | 0.003±0.001 | 0.004±0.004 | 0.003±0.000 | 0.004±0.000 |
| H+P | 0.007±0.008 | 0.002±0.003 | 0.003±0.008 | 0.000±0.001 | 0.002±0.001 |
| H+R | 0.006±0.007 | 0.002±0.002 | 0.003±0.006 | 0.000±0.001 | 0.002±0.001 |
| P+R | 0.006±0.007 | 0.002±0.002 | 0.003±0.008 | 0.001±0.002 | 0.002±0.002 |
| H+P+R | 0.007±0.008 | 0.002±0.003 | 0.003±0.007 | 0.000±0.001 | 0.002±0.001 |
| **SI-SDR ↑** | | | | | |
| H | -38.31±11.16 | -29.57±10.69 | -31.44±12.68 | -29.02±13.35 | -35.13±10.96 |
| P | -37.84±11.40 | -30.23±10.97 | -31.54±12.47 | -28.22±14.50 | -37.92±12.14 |
| R | -35.39±11.62 | -42.88±12.24 | -42.47±13.37 | -47.06±13.03 | -39.90±11.08 |
| H+P | -38.09±11.35 | -28.48±10.85 | -29.61±11.81 | -27.37±13.05 | -35.60±10.38 |
| H+R | -36.73±10.26 | -31.49±10.97 | -32.59±12.22 | -30.31±12.97 | -34.57±11.40 |
| P+R | -36.36±9.88 | -32.03±10.49 | -33.50±12.33 | -31.40±14.22 | -36.15±11.63 |
| H+P+R | -37.76±11.22 | -29.29±10.81 | -30.30±11.50 | -27.84±13.22 | -35.33±10.77 |
| **ViSQOL ↑** | | | | | |
| H | 3.63±0.57 | 3.30±0.90 | 3.35±0.92 | 4.17±0.71 | 3.88±0.36 |
| P | 3.65±0.61 | 3.24±0.91 | 3.23±0.89 | 4.15±0.73 | 3.83±0.55 |
| R | 2.52±0.54 | 2.71±0.63 | 2.71±0.66 | 2.21±0.82 | 3.28±0.48 |
| H+P | 3.73±0.56 | 3.40±0.93 | 3.36±0.92 | 4.21±0.69 | 3.90±0.42 |
| H+R | 3.54±0.64 | 3.18±0.85 | 3.33±0.87 | 4.14±0.73 | 3.91±0.35 |
| P+R | 3.31±0.65 | 2.88±0.71 | 3.04±0.81 | 3.92±0.79 | 3.83±0.50 |
| H+P+R | 3.75±0.56 | 3.34±0.89 | 3.34±0.89 | 4.20±0.70 | 3.96±0.38 |

As stated in Section 4.2, HP-codec and HP-codecX are trained on audio samples recorded at 44.1 kHz, upsampled at 48 kHz. A part of the high frequency latent representation is then dedicated to encoding silence. This suboptimal setting arises from constraints imposed by the language model prediction. We outline a potential workaround in Appendix C.

# 6 CONCLUSION

We introduce HP-Codec, a multi-branch neural audio codec that produces a latent representation disentangled across frequency bands. This disentanglement is further enhanced through a Harmonic–Percussive decomposition, which strengthens inter-band coupling and facilitates prediction of high-frequency representations from their low-frequency counterparts. In this way, we restructure the codec architecture to naturally support the downstream task of bandwidth extension. Building upon this design, we propose HP-CodecX, a bandwidth extension model that integrates HP-Codec with an autoregressive Transformer-based language model. The Transformer mirrors the codec's architecture, enabling effective modeling of cross-band dependencies. Empirical results across multiple datasets demonstrate that HP-CodecX achieves state-of-the-art performance on both objective and subjective metrics, yielding more accurate high-frequency reconstruction than existing baselines.

## USE OF LLMs

During the preparation of this manuscript, the authors employed Large Language Models in a limited capacity, specifically for text reformulation and figure layout refinement. These uses were auxiliary and do not constitute a substantive contribution of the LLMs to the development of the scientific content of this work.

## REPRODUCIBILITY STATEMENT

To facilitate reproducibility, our experiments were conducted on publicly available or otherwise reproducible datasets, and our approach builds upon reproducible models. We provide detailed technical descriptions throughout the paper when necessary, and we plan to release our implementation on the companion website upon acceptance (`https://harmonic-percussive-bandwidth-extension.github.io/`).

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

## A  APPENDIX: FINER HP-CODEC ANALYSIS

In this work, we proposed a disentanglement strategy for the representations extracted by a neural audio codec. Our approach modifies the structure of each RVQ in the model and leverages a training procedure inspired by harmonic–percussive decomposition (Fitzgerald, 2010; Driedger et al., 2014) to enforce disentanglement. Within this framework, each RVQ section is designed to capture a distinct spectral property of the input: the harmonic section encodes harmonic components, the percussive section captures percussive events, and the residual section models information not explained by the other two.

Building on the harmonic–percussive decomposition algorithm, we consider three signal components: harmonic, percussive, and residual (the latter capturing information not explained by the first two). Based on this, we designed an experiment in which HP-codec was provided with four types of input: full signals (**Global**), harmonic components (**H**), percussive components (**P**), and residual components (**R**). Reconstructions were then evaluated under four corresponding decoding settings, where only the relevant RVQ sections (**Global**, **H**, **P**, or **R**) were used. This setup yields $4 \times 4 = 16$ input–reconstruction pairs.

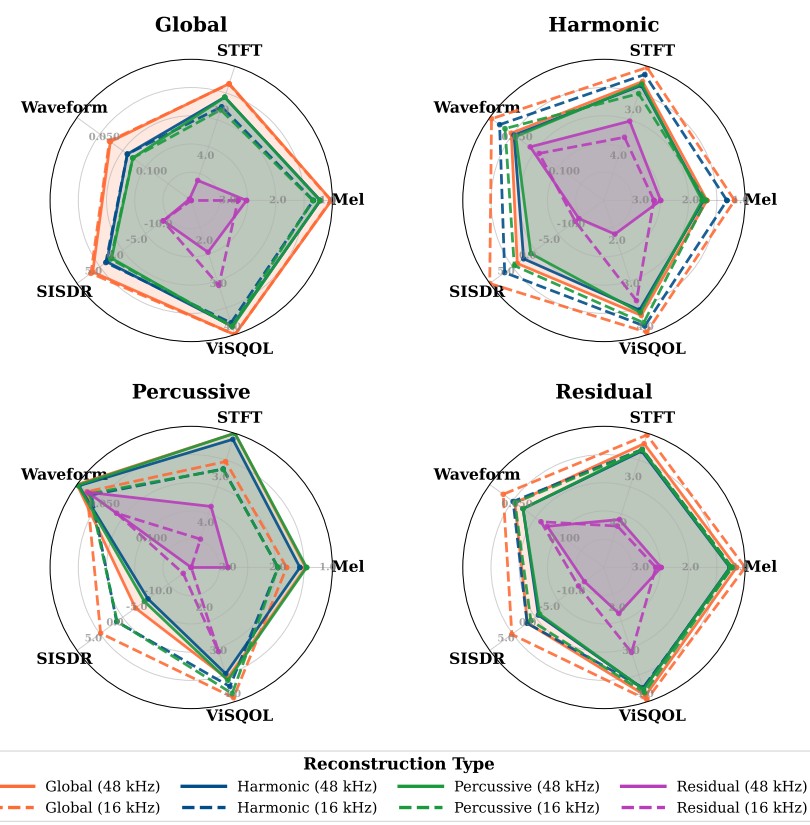

Figure 4: Reconstructions metrics of HP-codec, varying the spectral composition of the input (**Global**, **Harmonic**, **Percussive** or **Residual**), and the sections of the RVQs used for reconstruction. These graphs illustrate the values of Table 6.

The results of this experiment are reported in Fig. 4 and detailed in Table 6. Reconstructions using all sections (**Global**) consistently achieve the best performance across input types, indicating that the sections are complementary and each contributes information essential for accurate reconstruction. Reconstructions based solely on the residual section are consistently weaker than those using harmonic or percussive sections, suggesting that most of the signal information is effectively captured by harmonic and percussive components, as expected from harmonic–percussive decomposition.

Interestingly, percussive inputs reconstructed with the global configuration show limitations in the 16 kHz branch but are refined in the 48 kHz branch. For this type of input still, the percussive section

produces the most accurate reconstruction, whereas for harmonic inputs at 16 kHz, the harmonic section performs best. These findings align with the spectral distribution of natural audio: harmonic components dominate at low frequencies, while percussive components retain significant energy at higher frequencies. Overall, the results provide evidence that HP-codec achieves meaningful disentanglement of harmonic and percussive structures.

Table 6: Detailed reconstructions metrics ($\pm$ standard deviation) of HP-codec, varying the spectral composition of the input, and the sections of the RVQs used for reconstruction. From top to bottom, the first table gathers results for experiments where full signals were inputted, the second table is for harmonic parts of signals as input, the third for percussive parts and the last for residual parts. Each table is subdivised into sections, which correspond to the RVQ sections that were used for reconstruction.

| Input Signal | Used Sections | Sampling Rate | Mel ↓ | STFT ↓ | Waveform ↓ | SI-SDR ↑ | ViSQOL ↑ |
|---|---|---|---|---|---|---|---|
| Global | Global | 16000 | 0.80±0.08 | 2.30±0.29 | 0.051±0.015 | 6.74±2.53 | 4.33±0.09 |
| | | 48000 | 0.79±0.05 | 2.29±0.29 | 0.052±0.015 | 6.30±2.51 | 4.33±0.14 |
| | H | 16000 | 1.17±0.10 | 2.82±0.36 | 0.071±0.018 | 3.13±2.58 | 4.04±0.14 |
| | | 48000 | 1.02±0.08 | 2.60±0.35 | 0.071±0.018 | 2.88±2.58 | 4.12±0.15 |
| | P | 16000 | 1.18±0.18 | 2.89±0.41 | 0.077±0.020 | 1.95±2.81 | 4.12±0.11 |
| | | 48000 | 1.03±0.08 | 2.60±0.34 | 0.077±0.020 | 1.71±2.80 | 4.14±0.16 |
| | R | 16000 | 2.75±0.62 | 4.97±0.74 | 0.142±0.025 | -10.77±4.19 | 3.13±0.30 |
| | | 48000 | 2.57±0.60 | 4.51±0.63 | 0.140±0.025 | -10.83±4.19 | 2.32±0.41 |
| H | Global | 16000 | 0.99±0.18 | 1.92±0.29 | 0.017±0.005 | 10.11±2.37 | 4.27±0.12 |
| | | 48000 | 1.57±0.56 | 2.24±0.51 | 0.039±0.016 | 3.43±2.91 | 3.86±0.28 |
| | H | 16000 | 1.15±0.17 | 2.08±0.28 | 0.026±0.008 | 6.48±2.66 | 4.12±0.14 |
| | | 48000 | 1.64±0.56 | 2.33±0.50 | 0.042±0.015 | 1.88±3.08 | 3.73±0.30 |
| | P | 16000 | 1.59±0.36 | 2.52±0.57 | 0.032±0.012 | 4.15±3.60 | 4.04±0.16 |
| | | 48000 | 1.67±0.51 | 2.30±0.47 | 0.044±0.015 | 0.13±3.64 | 3.79±0.24 |
| | R | 16000 | 2.68±0.87 | 3.52±0.96 | 0.07±0.028 | -10.64±6.36 | 3.50±0.44 |
| | | 48000 | 2.54±0.80 | 3.15±0.64 | 0.060±0.019 | -11.55±7.09 | 1.88±0.51 |
| P | Global | 16000 | 1.73±0.69 | 2.54±0.75 | 0.026±0.012 | 4.38±3.17 | 4.23±0.11 |
| | | 48000 | 1.33±0.22 | 1.89±0.19 | 0.015±0.007 | -4.07±3.45 | 3.77±0.32 |
| | H | 16000 | 1.91±0.60 | 2.71±0.66 | 0.031±0.012 | 0.49±3.49 | 3.96±0.13 |
| | | 32000 | 1.45±0.24 | 2.03±0.22 | 0.017±0.008 | -7.11±4.04 | 3.65±0.32 |
| | P | 16000 | 1.91±0.71 | 2.72±0.79 | 0.029±0.012 | 0.39±3.84 | 4.13±0.11 |
| | | 48000 | 1.31±0.19 | 1.89±0.18 | 0.016±0.007 | -6.24±4.20 | 3.81±0.29 |
| | R | 16000 | 3.73±1.01 | 4.32±1.05 | 0.059±0.015 | -15.65±4.14 | 3.11±0.35 |
| | | 48000 | 2.96±0.80 | 3.57±0.66 | 0.027±0.015 | -17.61±3.25 | 1.06±0.30 |
| R | Global | 16000 | 0.77±0.09 | 1.93±0.22 | 0.030±0.012 | 4.74±2.85 | 4.28±0.085 |
| | | 48000 | 0.95±0.15 | 2.13±0.21 | 0.043±0.011 | 0.96±2.24 | 4.18±0.24 |
| | H | 16000 | 1.05±0.09 | 2.26±0.27 | 0.041±0.014 | 1.03±2.79 | 3.99±0.13 |
| | | 48000 | 1.09±0.14 | 2.30±0.25 | 0.052±0.013 | -1.71±2.34 | 4.02±0.22 |
| | P | 16000 | 1.04±0.13 | 2.27±0.28 | 0.043±0.015 | 0.12±3.10 | 4.11±0.08 |
| | | 48000 | 1.08±0.14 | 2.27±0.24 | 0.052±0.013 | -2.01±2.59 | 4.03±0.24 |
| | R | 16000 | 2.64±0.55 | 4.02±0.65 | 0.072±0.018 | -11.43±3.80 | 3.13±0.36 |
| | | 48000 | 2.53±0.53 | 3.87±0.60 | 0.079±0.017 | -12.88±3.62 | 2.18±0.58 |

## B  APPENDIX: FINER HP-CODECX ANALYSIS

Fig. 5 and Table 7 report the complete out-of-domain evaluation introduced in Section 4.7. We compare HP-codecX with two baselines across five test datasets: ENST-Drums, Medley-solos-DB, OrchideaSOL, Monophonic, and Polyphonic. Objective reconstruction metrics were computed on full-band signals (**Global**), low-frequency bands $[0\ \mathrm{kHz}, 8\ \mathrm{kHz}]$ (**LF**), and high-frequency bands $[8\ \mathrm{kHz}, 24\ \mathrm{kHz}]$ (**HF**). Full-band results were already summarized in Fig. 3.

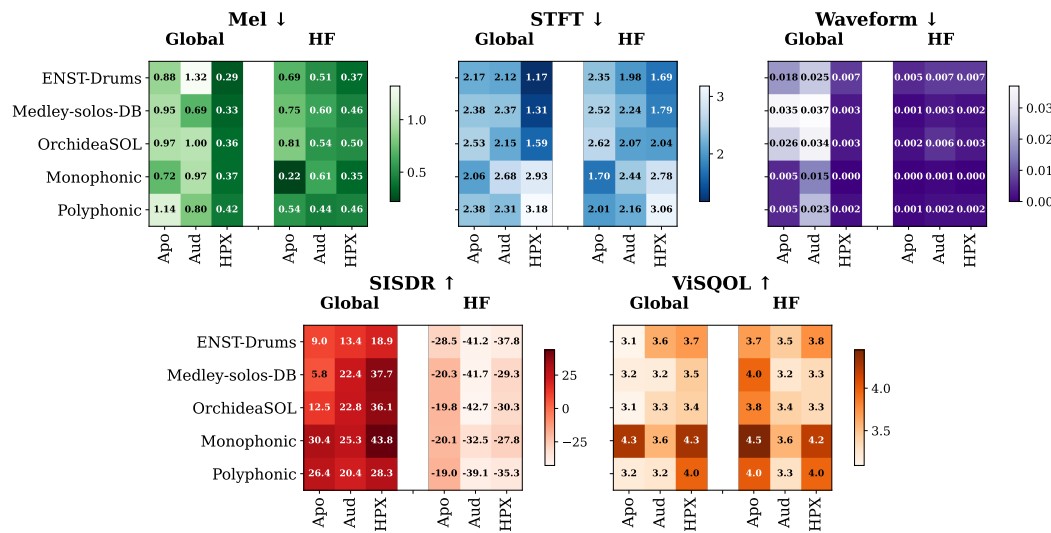

Figure 5: Objective reconstruction metrics, calculated on whole estimated signals (**Global**) and high frequency bands of estimated signals (**HF**). These metrics have been computed on Out-of-Domain test datasets: ENST-Drums, Medley-solos-DB, OrchideaSOL, Monophonic and Polyphonic. The Apollo (Apo) metrics are calculated at 44.1 kHz, while the AudioSR (Aud) and HP-codecX (HPX) metrics have been calculated at 48 kHz. These graphs illustrate the values contained in the Global and *HF* rows of Table 7.

As expected, **LF** scores serve mainly as reference since Apollo is the only model that fully reconstructs low frequencies. However, **HF** results reveal a clear spectral advantage of HP-codecX on percussive sources (ENST-Drums) and more general music signals (Medley-solos-DB, OrchideaSOL). This advantage diminishes for purely harmonic signals (Monophonic, Polyphonic), although our model remains competitive overall. Consistent with the objective results, informal listening tests confirmed that HP-codecX excels at reconstructing percussive and noise-like components relative to the baselines.

Table 7: Objective reconstruction metrics ($\pm$ standard deviation), calculated on whole estimated signals (**Global**), low frequency bands (**LF**) and high frequency bands (**HF**) of estimated signals. These metrics have been computed on Out-of-Domain test datasets: ENST-Drums, Medley-solos-DB, OrchideaSOL, Monophonic and Polyphonic. The Apollo metrics are calculated at 44.1 kHz, while the other metrics have been calculated at 48 kHz.

| | | ENST-drums | | | Monophonic | | |
|---|---|---|---|---|---|---|---|
| | | Apollo | AudioSR | HP-codecX | Apollo | AudioSR | HP-codecX |
| Mel ↓ | Global | 0.88±0.02 | 1.32±0.51 | **0.29**±0.16 | 0.72±0.45 | 0.97±0.64 | **0.37**±0.22 |
| | LF | 0.38±0.19 | 0.94±0.39 | 0.09±0.13 | 0.41±0.24 | 0.50±0.37 | 0.07±0.02 |
| | HF | 0.69±0.28 | 0.51±0.23 | 0.37±0.15 | 0.22±0.20 | 0.61±0.48 | 0.35±0.25 |
| STFT ↓ | Global | 2.17±0.57 | 2.12±0.72 | **1.17**±0.27 | **2.06**±0.32 | 2.68±1.45 | 2.93±0.67 |
| | LF | 0.47±0.18 | 0.81±0.36 | 0.14±0.09 | 0.79±0.11 | 0.80±0.45 | 0.71±0.27 |
| | HF | 2.35±0.71 | 1.98±0.70 | 1.69±0.45 | 1.70±0.27 | 2.44±1.37 | 2.78±0.25 |
| Waveform ↓ | Global | 0.018±0.019 | 0.025±0.018 | **0.007**±0.007 | 0.005±0.008 | 0.015±0.012 | **0.000**±0.001 |
| | LF | 0.015±0.018 | 0.022±0.018 | 0.001±0.002 | 0.005±0.008 | 0.015±0.012 | 0.000±0.000 |
| | HF | 0.005±0.006 | 0.007±0.007 | 0.007±0.008 | 0.000±0.000 | 0.001±0.001 | 0.000±0.001 |
| SI-SDR ↑ | Global | 8.97±12.45 | 13.43±10.95 | **18.90**±13.36 | 30.45±6.83 | 25.27±4.81 | **43.82**±14.12 |
| | LF | 13.31±14.17 | 23.85±6.83 | 37.82±9.50 | 31.50±6.51 | 27.19±4.51 | 56.53±8.99 |
| | HF | -28.49±8.41 | -41.22±12.21 | -37.76±11.22 | -20.10±13.44 | -32.45±14.40 | -27.84±13.22 |
| ViSQOL ↑ | Global | 3.09±0.87 | 3.60±0.59 | **3.70**±0.58 | 4.26±0.65 | 3.57±0.88 | **4.28**±0.55 |
| | LF | 4.61±0.10 | 4.59±0.12 | 4.70±0.03 | 4.67±0.05 | 4.36±0.34 | 4.72±0.02 |
| | HF | 3.69±0.84 | 3.45±0.68 | 3.75±0.56 | 4.45±0.43 | 3.57±0.96 | 4.20±0.70 |

| | | Medley-solos-DB | | | Polyphonic | | |
|---|---|---|---|---|---|---|---|
| | | Apollo | AudioSR | HP-codecX | Apollo | AudioSR | HP-codecX |
| Mel ↓ | Global | 0.95±0.20 | 0.69±0.30 | **0.33**±0.14 | 1.14±0.36 | 0.80±0.26 | **0.42**±0.10 |
| | LF | 0.34±0.08 | 1.10±0.42 | 0.03±0.01 | 0.55±0.17 | 0.50±0.20 | 0.08±0.01 |
| | HF | 0.75±0.25 | 0.60±0.24 | 0.46±0.18 | 0.54±0.22 | 0.44±0.16 | 0.46±0.14 |
| STFT ↓ | Global | 2.38±0.49 | 2.37±0.74 | **1.31**±0.35 | 2.38±0.22 | **2.31**±0.70 | 3.18±0.25 |
| | LF | 0.51±0.09 | 0.88±0.31 | 0.10±0.05 | 0.78±0.06 | 0.74±0.26 | 0.57±0.21 |
| | HF | 2.52±0.51 | 2.24±0.72 | 1.79±0.45 | 2.01±0.18 | 2.16±0.70 | 3.06±0.24 |
| Waveform ↓ | Global | 0.035±0.016 | 0.037±0.017 | **0.003**±0.006 | 0.005±0.005 | 0.023±0.010 | **0.002**±0.001 |
| | LF | 0.035±0.017 | 0.036±0.017 | 0.001±0.005 | 0.005±0.005 | 0.022±0.010 | 0.000±0.000 |
| | HF | 0.001±0.002 | 0.003±0.005 | 0.002±0.003 | 0.001±0.000 | 0.002±0.001 | 0.002±0.001 |
| SI-SDR ↑ | Global | 5.76±9.12 | 22.38±6.24 | **37.66**±12.00 | 26.37±4.17 | 20.45±4.08 | **28.30**±3.64 |
| | LF | 5.95±9.45 | 24.90±5.13 | 48.26±12.08 | 28.65±4.39 | 24.43±4.5 | 51.08±5.32 |
| | HF | -20.29±9.46 | -41.70±11.55 | -29.29±10.81 | -19.00±10.05 | -39.06±10.39 | -35.33±10.77 |
| ViSQOL ↑ | Global | 3.17±0.67 | 3.19±0.62 | **3.45**±0.66 | 3.20±0.58 | 3.25±0.58 | **3.97**±0.34 |
| | LF | 4.57±0.13 | 4.40±0.22 | 4.70±0.03 | 4.65±0.04 | 4.34±0.21 | 4.71±0.03 |
| | HF | 3.96±0.70 | 3.25±0.65 | 3.34±0.89 | 4.01±0.42 | 3.29±0.56 | 3.96±0.38 |

| | | OrchideaSOL | | | - | | |
|---|---|---|---|---|---|---|---|
| | | Apollo | AudioSR | HP-codecX | - | - | - |
| Mel ↓ | Global | 0.97±0.32 | 1.00±0.65 | **0.36**±0.23 | - | - | - |
| | LF | 0.34±0.13 | 0.63±0.47 | 0.04±0.03 | - | - | - |
| | HF | 0.81±0.37 | 0.54±0.31 | 0.50±0.33 | - | - | - |
| STFT ↓ | Global | 2.53±0.60 | 2.15±0.93 | **1.59**±0.71 | - | - | - |
| | LF | 0.52±0.13 | 0.80±0.40 | 0.19±0.21 | - | - | - |
| | HF | 2.62±0.76 | 2.07±0.85 | 2.04±0.78 | - | - | - |
| Waveform ↓ | Global | 0.026±0.020 | 0.034±0.026 | **0.003**±0.007 | - | - | - |
| | LF | 0.025±0.019 | 0.032±0.026 | 0.000±0.001 | - | - | - |
| | HF | 0.002±0.004 | 0.006±0.011 | 0.003±0.007 | - | - | - |
| SI-SDR ↑ | Global | 12.52±12.03 | 22.80±10.01 | **36.09**±13.38 | - | - | - |
| | LF | 13.10±12.48 | 26.79±7.67 | 49.30±12.05 | - | - | - |
| | HF | -19.75±9.04 | -42.73±13.62 | -30.30±11.50 | - | - | - |
| ViSQOL ↑ | Global | 3.10±0.87 | 3.34±0.79 | **3.42**±0.79 | - | - | - |
| | LF | 4.60±0.14 | 4.46±0.29 | 4.71±0.04 | - | - | - |
| | HF | 3.82±0.89 | 3.41±0.87 | 3.34±0.89 | - | - | - |

# C APPENDIX: A 32 KHZ MODEL

A limitation of HP-codec and HP-codecX arises from the mismatch between the operating sampling rate of the codec (48 kHz) and the recording rate of the training data (44.1 kHz). As a result, the 48 kHz branch cannot be fully exploited: part of its capacity is used to encode silence in the $[22.05 \text{ kHz}, 24 \text{ kHz}]$ band. This issue stems from the requirement that both branches of HP-codec yield the same number of tokens per input, ensuring compatibility with the language model. Consequently, the high-frequency branch sampling rate must be an integer multiple of that of the low-frequency branch.

To address this, we reconfigured the system to perform bandwidth extension from 16 kHz to 32 kHz. We trained HP-codec32, a derived version of HP-codec, modifying only the encoder ratios of the high-frequency branch to $\{2, 2, 5, 8\}$, while keeping the language model unchanged (forming HP-codec32X), and subsequently evaluated its performance.

Table 8: Objective reconstruction metrics ($\pm$ standard deviation) for the Apollo (44.1 kHz), AudioSR (48 kHz) models and HP-codec32X (32 kHz). The top metrics are calculated over the whole signals (**Global**). The lower metrics are calculated on $[8 \text{ kHz}, 22.05 \text{ kHz}]$ bands (**HF**). For a fair comparison, we upsampled the results of our model to the sampling rate of the model we want to compare to.

| **Global** | Apollo (Li & Luo, 2025) | HP-codec32X (ups. at 44.1 kHz) | AudioSR (Liu et al., 2024a) | HP-codec32X (ups. at 48 kHz) |
|---|---|---|---|---|
| Mel $\downarrow$ | $1.02 \pm 0.12$ | $\mathbf{0.24} \pm 0.06$ | $1.83 \pm 0.43$ | $\mathbf{0.23} \pm 0.061$ |
| STFT $\downarrow$ | $3.35 \pm 0.53$ | $\mathbf{1.77} \pm 0.17$ | $3.98 \pm 0.71$ | $\mathbf{1.96} \pm 0.15$ |
| Waveform $\downarrow$ | $0.048 \pm 0.013$ | $\mathbf{0.011} \pm 0.006$ | $0.069 \pm 0.019$ | $\mathbf{0.011} \pm 0.006$ |
| SI-SDR $\uparrow$ | $3.26 \pm 3.82$ | $\mathbf{20.81} \pm 7.24$ | $13.92 \pm 5.06$ | $\mathbf{20.81} \pm 7.24$ |
| ViSQOL $\uparrow$ | $3.26 \pm 0.40$ | $\mathbf{3.67} \pm 0.31$ | $2.98 \pm 0.37$ | $\mathbf{3.67} \pm 0.31$ |
| **HF** | Apollo (Li & Luo, 2025) | HP-codec32X (ups. at 44.1 kHz) | AudioSR (Liu et al., 2024a) | HP-codec32X (ups. at 48 kHz) |
| Mel $\downarrow$ | $0.80 \pm 0.12$ | $\mathbf{0.26} \pm 0.06$ | $0.83 \pm 0.16$ | $\mathbf{0.26} \pm 0.06$ |
| STFT $\downarrow$ | $3.04 \pm 0.51$ | $\mathbf{1.88} \pm 0.14$ | $3.09 \pm 0.54$ | $\mathbf{2.06} \pm 0.14$ |
| Waveform $\downarrow$ | $\mathbf{0.008} \pm 0.005$ | $0.011 \pm 0.006$ | $\mathbf{0.010} \pm 0.005$ | $0.011 \pm 0.006$ |
| SI-SDR $\uparrow$ | $\mathbf{-28.39} \pm 8.42$ | $-34.22 \pm 8.03$ | $-44.20 \pm 10.18$ | $\mathbf{-34.22} \pm 8.04$ |
| ViSQOL $\uparrow$ | $3.38 \pm 0.37$ | $\mathbf{3.80} \pm 0.30$ | $3.18 \pm 0.35$ | $\mathbf{3.80} \pm 0.30$ |

Table 8 reports objective reconstruction metrics computed on full signals (**Global**) and on the high-frequency band (**HF**). For fair comparison, our outputs were upsampled to match the sampling rates of the baseline models (44.1 kHz for Apollo and 48 kHz for AudioSR). Consistent with the 48 kHz setting, HP-codec32X achieves superior spectral reconstruction compared to both baselines. Although SI-SDR indicates higher distortion in the high-frequency range, informal listening tests suggest that these distortions are not perceptually salient, further underscoring the limitations of SI-SDR as a metric for synthesis tasks.

Comparing Table 8 with Table 2, we find that both HP-codec32X and HP-codecX achieve similarly strong reconstruction metrics. This confirms that training the 48 kHz model on data recorded at 44.1 kHz does not lead to a loss of efficiency.

## D  APPENDIX: SPECTROGRAMS OF EXTENDED SIGNALS

In Fig. 6 and Fig. 7 we display the spectrograms of the 12 signals drawn from the MUSDB18 test set for the MUSHRA evaluation introduced in Section 4.4, in four different settings: a reference spectrogram, a spectrogram of the estimation drawn from the Apollo model, one from the AudioSR model and finally HP-codecX estimation.

The spectrogram analysis highlights several characteristics of HP-codecX. First, it provides more accurate percussive reconstructions (visible through the vertical structures of the spectrograms) than the baselines, as observed in samples 11, 105, 131, 535, and 792. Second, it generates denser high-frequency estimates, as illustrated in samples 189, 658, and 792. However, this sometimes leads to artifacts in the high-frequency range, where the model attempts to reconstruct content absent from the reference (sample 723). Interestingly, there are also cases where the spectrogram suggests poor estimation (sample 407), while listening tests confirm that the output remains perceptually satisfactory due to the low energy in the affected bands.

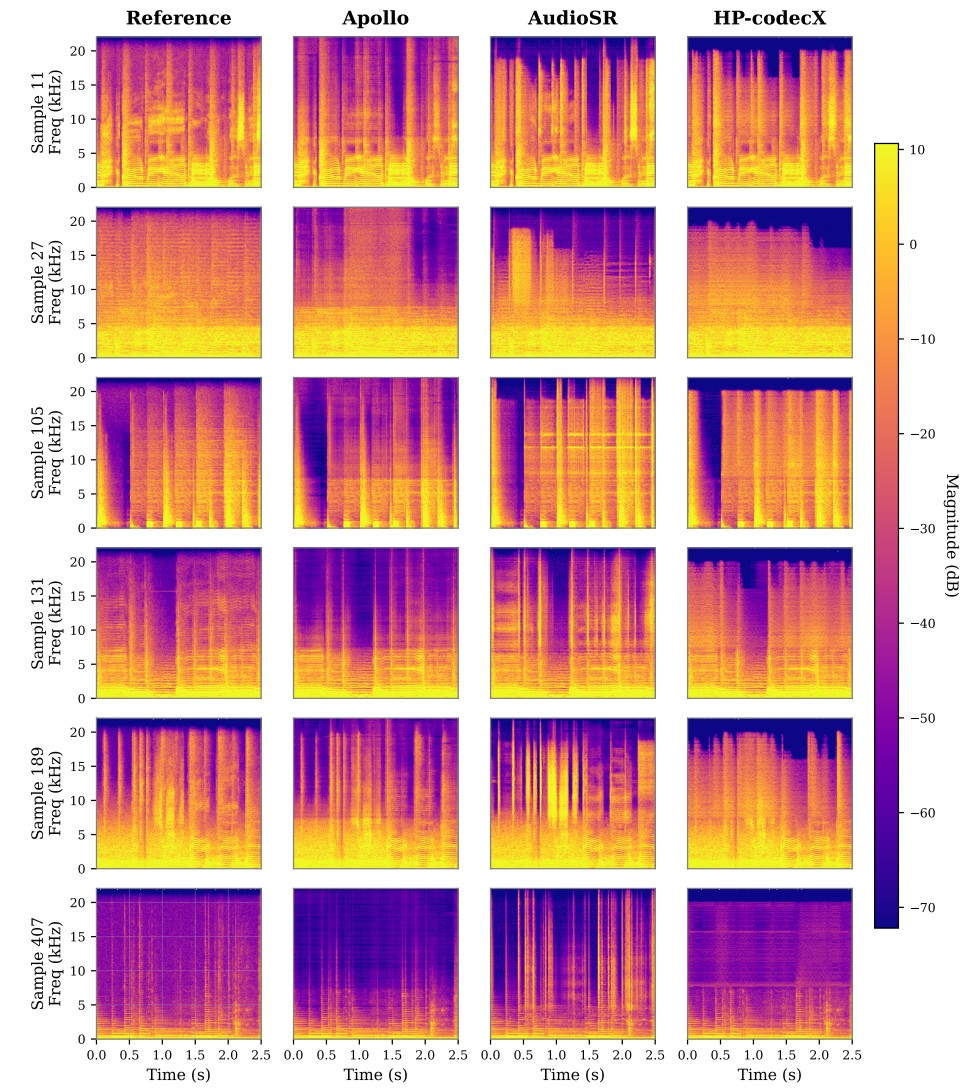

Figure 6: Spectrograms of estimated signal drawn from Apollo, AudioSR and HP-codecX (1/2)

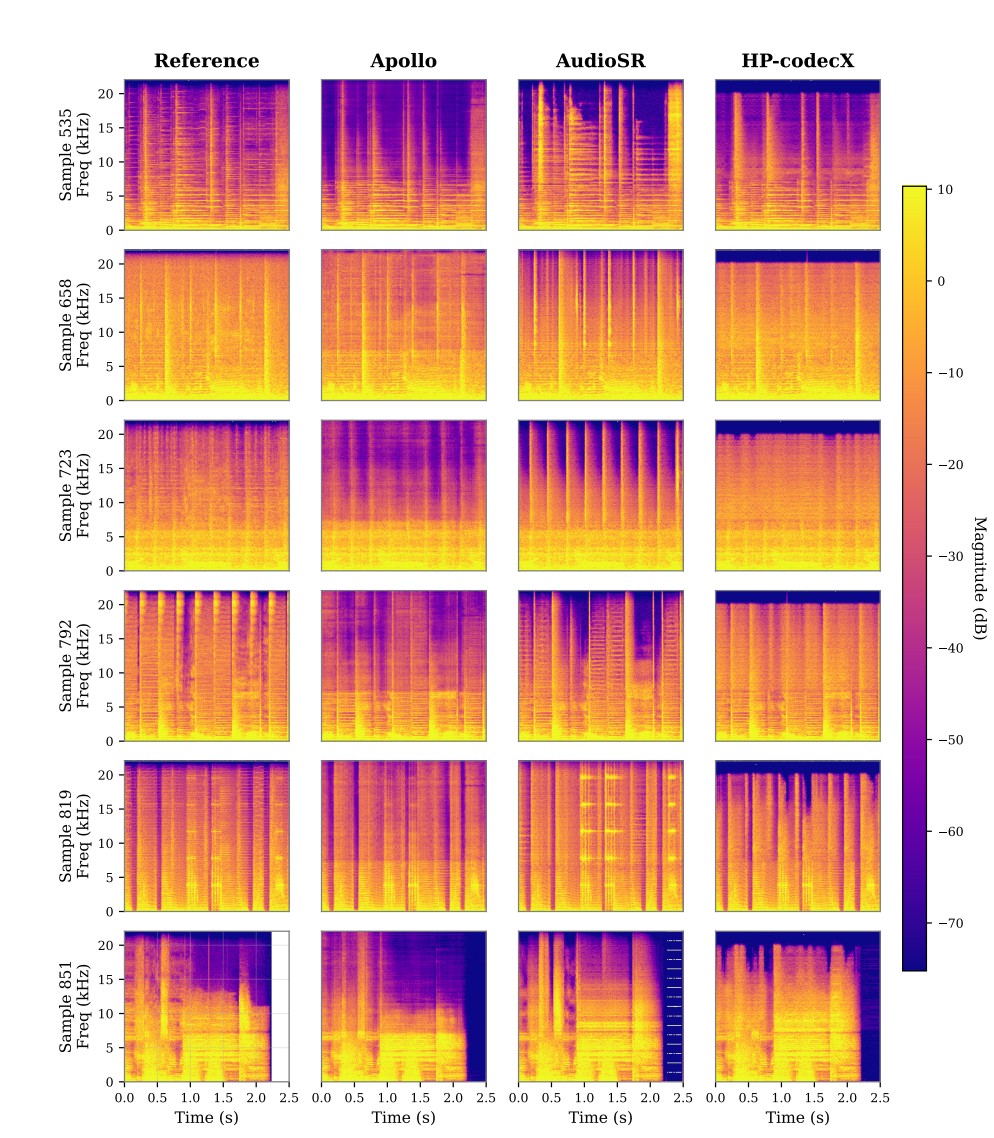

Figure 7: Spectrograms of estimated signal drawn from Apollo, AudioSR and HP-codecX (2/2)

