# OpenReview forum: "Harmonic-Percussive Disentangled Neural Audio Codec for Bandwidth Extension"
_ICLR.cc/2026/Conference — Submitted to ICLR 2026_

### Official Review · Reviewer_jmoQ · 2025-10-31

**Soundness:** 2
**Presentation:** 2
**Contribution:** 2
**Rating:** 4
**Confidence:** 3

**Summary:**

This paper proposes HP-codecX, a novel two-stage approach for audio bandwidth extension. The method first employs a neural audio codec, HP-codec, which is designed to produce a disentangled latent representation. This disentanglement is structured along two axes: separating low and high frequency bands, and further decomposing each band's representation into harmonic, percussive, and residual components. In the second stage, a multi-branch Transformer-based language model is trained on these discrete tokens to predict the high-frequency tokens from the low-frequency ones. The authors present extensive experiments, including objective metrics and subjective MUSHRA listening tests, demonstrating that their method achieves state-of-the-art performance compared to recent strong baselines like Apollo and AudioSR.

**Strengths:**

The idea of structuring the latent space of a neural audio codec to explicitly align with a downstream task is a very strong and promising direction. The authors' design, which introduces a harmonic-percussive decomposition as an inductive bias, is well-motivated by principles of audio signal processing and aims to create a more predictable and interpretable representation for the language model. The empirical results presented are a clear strength of this work. The proposed HP-codecX consistently outperforms strong, recent baselines on both in-domain and out-of-domain datasets, as evidenced by a comprehensive suite of objective metrics and, importantly, a well-conducted MUSHRA listening test. The thoroughness of the evaluation adds significant credibility to the paper's claims of state-of-the-art performance.

**Weaknesses:**

While the results are impressive, I find the paper's fundamental premise to be insufficiently justified, which constitutes a significant weakness. The entire approach is built on using a neural audio codec, which is an inherently lossy compression process. The paper does not adequately explain why introducing this information bottleneck is a desirable step for a high-fidelity restoration task like bandwidth extension. It seems counter-intuitive to first discard information through compression, only to then try and hallucinate even more information (the high frequencies). The advantage of this token-based prediction over more direct spectrogram- or waveform-based inpainting/generation methods is not made clear.
This motivational weakness is further exacerbated by a critical lack of detail regarding the cornerstone of the method: the harmonic-percussive-residual decomposition. The paper states that the training is guided by this decomposition, but it never specifies how this decomposition is performed. Is it a classic algorithm like median filtering? Is it a learned component? How are the decomposed signals used to train the respective "harmonic," "percussive," and "residual" RVQ streams? Without these crucial details, the central claim of achieving "semantically informed disentanglement" is unsubstantiated and the method is not fully reproducible.

**Questions:**

To better understand the contributions and rationale of your work, I would appreciate clarification on the following points:
1. Could you please elaborate on the fundamental advantage of using a lossy codec as a pre-processor for bandwidth extension? Why is a compressed, discrete token representation a better starting point for this task than the original, continuous low-band spectrogram or waveform, which contains more pristine information?
2. For the training of the HP-codec, you mention specific iterations for harmonic and percussive components. Could you provide the specific details of the harmonic-percussive-residual decomposition algorithm used to generate the training data for these steps? This information is essential for understanding your method and for reproducibility.
3. Following up on the above, the ablation in Table 5 is interesting. It shows that the harmonic and percussive streams are indeed better at reconstructing their respective signal types. However, given the unclear decomposition process, could this result simply be a product of the training data curation? More importantly, what happens to the overall bandwidth extension quality if you ablate one of the streams (e.g., predict only with harmonic and residual tokens)? This would help quantify the actual contribution of each "semantic" branch to the downstream task.

---

> ### Author Response · Authors · 2025-11-20
>
> # Could you please elaborate on the fundamental advantage of using a lossy codec as a pre-processor for bandwidth extension?
>
> The reviewer raises a valid concern regarding our choice to perform bandwidth extension directly from the compressed latent representation of a neural audio codec. Intuitively, discarding part of the original information could increase the difficulty of the task or introduce hallucination. While this intuition is reasonable, several recent synthesis frameworks, including VALL-E (Wang et al., 2023), which directly motivates our approach, demonstrate that operating on discrete codec tokens can in fact improve generative performance.
>
> The underlying rationale is that transformer-based generative models benefit from operating on simplified, discrete token sequences that resemble textual data, enabling NLP-style modeling. A neural audio codec provides a compact and discrete representation that is well aligned with this requirement. Although compressing the input inevitably removes detail, retaining too much low-level information can actually hinder the model by increasing complexity and encouraging overfitting to irrelevant structure, which may itself lead to hallucinations or artifacts. By filtering out redundant details, the codec representation creates a more tractable modeling space for the bandwidth-extension transformer.
>
> We acknowledge that these motivations were not clearly articulated in the submitted version and we will clarify and expand this discussion in the revised manuscript.
>
> ---
>
> # Could you provide the specific details of the harmonic-percussive-residual decomposition algorithm used to generate the training data for these steps?
>
> We also agree that the presentation of the harmonic–percussive–residual decomposition was insufficiently explicit. The method was mentioned only briefly in Section 3.1.3 (“with inputs derived from harmonic–percussive–residual decomposition (Driedger et al., 2014)”), which may have caused confusion. In the revised version, we will introduce this algorithm more clearly and include a short description of the decomposition procedure.
>
> ---
>
> # More importantly, what happens to the overall bandwidth extension quality if you ablate one of the streams (e.g., predict only with harmonic and residual tokens)?
>
> We thank the reviewer for noting that the motivation and impact of the semantic sections could be explained more clearly. To clarify their contribution, we conducted additional experiments, including an ablation study where we remove each semantic section independently to measure its effect on reconstruction quality. These new results (to appear in the revised manuscript) show the distinct influence of each component. For instance, datasets with purely harmonic content are better reconstructed by the sole harmonic section. (For brevity in the rebuttal, we report only Mel and ViSQOL metrics below; the full set of metrics will be included in the revised version)
>
> ---
>
> ## Objective metrics (± standard deviation) across datasets, evaluated on high-frequency bands.
> We display estimations based on harmonic (H), percussive (P), or residual (R) tokens,
> and on combinations (H+P, H+R, P+R, H+P+R).
>
> | Dataset | Metric | H | P | R | H+P | H+R | P+R | H+P+R |
> |--------|--------|---|---|---|-----|-----|-----|--------|
> | **ENST-drums** | Mel ↓ | 0.41±0.16 | 0.40±0.15 | 0.89±0.29 | 0.41±0.16 | 0.38±0.14 | 0.39±0.13 | 0.37±0.15 |
> | | ViSQOL ↑ | 3.63±0.57 | 3.65±0.61 | 2.52±0.54 | 3.73±0.56 | 3.54±0.64 | 3.31±0.65 | 3.75±0.56 |
> | **Medley-solos-DB** | Mel ↓ | 0.51±0.20 | 0.48±0.19 | 0.90±0.32 | 0.50±0.20 | 0.47±0.17 | 0.46±0.15 | 0.46±0.18 |
> | | ViSQOL ↑ | 3.30±0.90 | 3.24±0.91 | 2.71±0.63 | 3.40±0.93 | 3.18±0.85 | 2.88±0.71 | 3.34±0.89 |
> | **OrchideaSOL** | Mel ↓ | 0.53±0.35 | 0.54±0.35 | 0.83±0.48 | 0.55±0.35 | 0.47±0.28 | 0.49±0.28 | 0.50±0.33 |
> | | ViSQOL ↑ | 3.35±0.92 | 3.23±0.89 | 2.71±0.66 | 3.36±0.92 | 3.33±0.87 | 3.04±0.81 | 3.34±0.89 |
> | **Monophonic** | Mel ↓ | 0.33±0.25 | 0.37±0.28 | 1.67±0.22 | 0.36±0.24 | 0.40±0.25 | 0.46±0.27 | 0.35±0.25 |
> | | ViSQOL ↑ | 4.17±0.71 | 4.15±0.73 | 2.21±0.82 | 4.21±0.69 | 4.14±0.73 | 3.92±0.79 | 4.20±0.70 |
> | **Polyphonic** | Mel ↓ | 0.44±0.14 | 0.52±0.18 | 1.23±0.25 | 0.49±0.14 | 0.47±0.12 | 0.53±0.15 | 0.46±0.14 |
> | | ViSQOL ↑ | 3.88±0.36 | 3.83±0.55 | 3.28±0.48 | 3.90±0.42 | 3.91±0.35 | 3.83±0.50 | 3.96±0.38 |

---

> > ### Author Response · Authors · 2025-11-26
> >
> > We thank the reviewer again for their helpful feedback. We have submitted a revised version of the paper in which all modifications are highlighted in red. Below, we summarize the changes directly addressing the reviewer’s comments.
> >
> > ---
> >
> > ### (1) Expanded explanation of our approach (Section 3)
> >
> >
> > Following the reviewer’s suggestion, we added a paragraph clarifying the motivation behind applying our discrete-token, transformer-based framework to bandwidth extension:
> >
> >
> > >“To this end, we leverage the generative modeling capabilities of transformer-based language models by operating in a discrete token space. Neural audio codecs provide compact discrete representations that are well suited for such models, enabling the use of NLP-style sequence modeling techniques. While discretization necessarily discards some fine-grained information, it also removes low-level variability that can complicate learning, reduce generalization, or induce artifacts. In practice, the codec representation yields a cleaner and more tractable modeling domain in which the transformer can focus on predicting the missing high-frequency structure.”
> >
> > ---
> >
> > ### (2) Clarification of the semantic decomposition method (Section 4.2)
> >
> >
> > In line with the reviewer’s request, we now explicitly state that we use the algorithm of Driedger et al. (2014) and provide a brief explanation of its core mechanism:
> >
> >
> > >“The harmonic–percussive–residual decomposition used during training follows the procedure of Driedger et al. (2014). The method applies horizontal and vertical median filtering to the magnitude spectrogram, yielding estimates of the harmonic and percussive components, respectively. The residual component is then defined as the part of the signal not captured by either of these two estimates.”
> >
> > ---
> >
> > ### (3) Addition of decomposition results (Section 4.8)
> >
> >
> > As suggested by the reviewer, we included the new decomposition experiment, now presented in Table 5, and added the following commentary:
> >
> >
> > >“Additionally, comparing reconstructions obtained from different token combinations highlights how each section contributes to overall fidelity. For example, harmonic signals are best reconstructed using harmonic tokens alone.”
> >
> > ---
> >
> > Additionally, other edits were made throughout the manuscript to improve clarity and strengthen the justification of our design choices. We invite the reviewer to consult the revised version for the complete set of updates.

---

### Official Review · Reviewer_EHRK · 2025-10-31

**Soundness:** 3
**Presentation:** 3
**Contribution:** 2
**Rating:** 4
**Confidence:** 3

**Summary:**

The paper proposes HP-codecX, a bandwidth-extension framework built on a two-branch neural codec (at 16 kHz and 48 kHz) that internally disentangles tokens into harmonic, percussive, and residual sections. A lightweight token-level language model (implemented as three small transformer modules) predicts the missing 48 kHz tokens conditioned on the 16 kHz tokens. The 48 kHz decoder then reconstructs the wideband signal. The authors argue that the semantic disentangling improves modeling difficulty by routing different spectral/temporal structures into separate RVQ pathways. They show objective and subjective improvements over AudioSR and Apollo, provide qualitative evidence of specialization, and present a small MUSHRA-style listening test.

**Strengths:**

Conceptually aligned inductive bias. Splitting harmonic/percussive/residual structure matches known spectral decompositions in music/audio and is a reasonable prior for bandwidth extension.

* The model decomposition (two-branch codec -> semantic RVQ -> LM) is straightforward and can be done with standard components.

* Ablations on harmonic and percussive inputs show that the intended sections contribute as designed.

* The proposed method outperforms strong baselines on many objective metrics and is preferred in a small subjective test.

* Alternating specialization updates and staged training is practical and computationally moderate.

**Weaknesses:**

* The system is essentially a composition of existing components (RVQ codec + token LM).
The semantic split is a hand-crafted heuristic, not a new modeling paradigm, and the architecture is also taken from existing work. Novelty overall is therefore limited.

* One of the core claims regarding the necessity of the semantic split is not demonstrated. The ablation removes the semantic split but replaces three deeper transformers with a single-layer LM, and only makes very vague claims as to why ("Only the single-layer transformer was able to learn non-trivial representations"). As this is one of the core components that the paper claims is necessary, this ablation needs to be done properly and in a setting where the capacity of the ablation vs. the baseline is comparable. It is also possible that the split into more codebooks/tokens is what yields better performance and not necessarily that the split is done semantically.

* The paper does not evaluate against mmodels such as n Grumiaux & Lagrange  and it is not clear why. The paper apparently even uses parts from this paper for the data. This makes the claim of SOTA performance a bit questionable.

* Code is not provided (only “upon acceptance”), which makes it impossible to verify and understand all details.

**Questions:**

* Why was the non-semantic baseline restricted to a single transformer layer?

* Will you add a parameter-matched unified LM baseline to properly isolate semantic structure effects?

* Why were some other baselines omitted?

---

> ### Author Response · Authors · 2025-11-20
>
> # Why was the non-semantic baseline restricted to a single transformer layer ?
>
> The reviewer correctly points out that we did not fully justify the choice of a single-layer transformer as the non-semantic baseline, given that the semantic configuration uses three deep transformers. As noted briefly in the submitted paper, this single-layer model was the only one among a larger set of tested architectures, including deeper transformer variants, that did not collapse or nearly collapse during training (producing nearly identical tokens regardless of input). We hypothesize that the task complexity caused larger models to suffer from vanishing or uninformative gradients, which, when propagated through multiple layers, led to this collapse. In the revised version, we will provide additional context on this selection process and include a concise summary of the alternative architectures that failed to converge.
>
> ---
>
> # Will you add a parameter-matched unified LM baseline to properly isolate semantic structure effects ?
>
> To further substantiate the motivation behind our semantic design, we conducted additional experiments that remove the semantic decomposition while keeping the rest of the HP-codecX architecture unchanged. Specifically, we have evaluated a variant with three deep transformer models identical to those in HP-codecX but operating without any semantic decomposition, and a variant using a single transformer shared across the three RVQ branches. As suggested by the reviewer, these configurations allow us to isolate the effect of semantic decomposition. The results show that architectures identical in all other respects achieve better reconstruction performance when semantic decomposition is applied. Furthermore, they also demonstrate the benefit of using separate transformers rather than a single shared one. (Please see the results below).
>
> ---
>
> # Code is not provided (only “upon acceptance”), which makes it impossible to verify and understand all details.
>
> We acknowledge that providing the code during the review process would have simplified verification. However, because the submission and reviews are public, we preferred not to release the full implementation in advance to avoid the risk of unauthorized use. The code will be released upon acceptance.
>
> ---
>
> # Why were some other baselines omitted ?
>
> The reviewer also notes the absence of a comparison to Grumiaux & Lagrange (2023). This comparison would indeed have been natural, since we rely on part of their work for data generation and cite them in the literature review. It was originally planned, but the authors did not release a trained model, unlike the baselines of Liu et al. (2024) and Li & Luo (2025). Our attempts to train their system were unsuccessful because their preprocessing pipeline depends on a specific cuDNN version that we were unable to install on two different of our lab machines, and any workaround would have required intrusive and non-trivial modifications to their code. We will clarify this limitation more explicitly in the revised manuscript.
>
> ---
>
> ## Objective metrics (± standard deviation) for the bandwidth-extension task.
> Results are reported for:
> (i) **HP-codecX**, which uses three RVQ sections and three transformers;
> (ii) a model with three RVQ sections and three transformers of equivalent depth trained without semantic decomposition (**EXP1**);
> (iii) a model with three RVQ sections and a shared transformer trained without semantic decomposition (**EXP2**); and
> (iv) a model without semantic RVQ sections and a single-layer transformer (**EXP3**).
> Metrics are evaluated on the high-frequency (HF) band.
>
> | **HF**        | **HP-codecX**      | **EXP1**          | **EXP2**          | **EXP3**          |
> |---------------|---------------------|--------------------|--------------------|--------------------|
> | Mel ↓         | **0.49 ± 0.14**     | 0.78 ± 0.19        | 0.78 ± 0.18        | 0.62 ± 0.14        |
> | STFT ↓        | **2.06 ± 0.40**     | 2.74 ± 0.60        | 2.84 ± 0.56        | 2.29 ± 0.43        |
> | Waveform ↓    | **0.012 ± 0.007**   | 0.016 ± 0.012      | 0.017 ± 0.012      | 0.014 ± 0.009      |
> | SI-SDR ↑      | -36.82 ± 8.40       | **-34.92 ± 11.74** | -37.89 ± 10.36     | -37.30 ± 9.15      |
> | ViSQOL ↑      | **3.53 ± 0.46**     | 2.60 ± 0.58        | 2.84 ± 0.55        | 3.03 ± 0.45        |

---

> > ### Comment · Reviewer_EHRK · 2025-11-23
> >
> > Thank you for the rebuttal. I find it very surprising that only the single layer transformer was able to converge at all and this definitely needs further attention. You mention that you want to add more to the revised paper, but I'm not sure whether I'm just not able to find the paragraph for this or whether you did not update the paper yet.
> > The missing baseline from Grumiaux & Lagrange (2023) is also concerning and the claim that it didn't install due to one dependency is not very convincing. You mention a revised manuscript again, but I don't see anything regarding this.

---

> > > ### Author Response · Authors · 2025-11-24
> > >
> > > We thank the reviewer for the feedback. In response to the concerns regarding the motivation and validation of the semantic RVQ decomposition, we will revise Section 4.8 (“Validating the semantic sections”) to clarify the empirical evidence that led to the proposed design. The updated section will begin with the following paragraph:
> > >
> > > “Tables 4 and 5 highlight the role of the RVQs semantic sections. Using this decomposition of the data together with RVQs was initially motivated by a series of negative experimental results. Basing the prediction on the architecture proposed in Giniès et al. (2025), we trained multiple transformer models to predict high-frequency tokens from their low-frequency counterparts. We conducted nine training runs, varying the model depth from 1 to 15 layers and the input duration from 0.33 s to 5 s. Most configurations collapsed, producing erroneous predictions restricted to a small subset of tokens. Even simplified variants of the task, where the diversity of input tokens was reduced by half, failed to converge. Among all experiments, only shallow models exhibited any degree of successful learning, with the single-layer transformer achieving the most reliable training behavior. These results suggest that the task complexity induces weak gradient signals which, when propagated through deeper architectures, lead to systematic collapse.
> > >
> > > To further validate the choice of semantic decomposition and the split-transformer design, we trained two additional models that omit semantic partitioning of the training data: one using three deep transformers identical to those in HP-codecX, and another using a single transformer shared across the three RVQ branches. As shown in Table 4, both variants and the single-layer reference introduced earlier perform markedly worse than HP-codecX, confirming the advantage of incorporating explicit semantic sections.”
> > >
> > > ---
> > >
> > > Concerning the Grumiaux & Lagrange baseline, we would like to underline that the authors’ implementation is limited to bandwidth extension from 4 kHz to 16 kHz, which prevents a meaningful comparison with our model in the conditions used in this paper. In spite of the several hours we invested, attempting to run their code, we could not use their model in a way that would yield informative results in our context.

---

> > > > ### Author Response · Authors · 2025-11-26
> > > >
> > > > We thank the reviewer again for their helpful feedback. We have submitted a revised version of the paper in which all modifications are highlighted in red. Below, we summarize the changes directly addressing the reviewer’s comments.
> > > >
> > > > ---
> > > >
> > > > ### (1) Clarification of the single-layer transformer experiment and introduction of new results supporting semantic decomposition (Section 4.8)
> > > >
> > > >
> > > > To clarify the motivation behind semantic decomposition and to address the reviewer’s request for additional evidence, we added two paragraphs to Section 4.8.
> > > >
> > > >
> > > > First, we expanded our explanation of the experimental trajectory that led to the introduction of semantic decomposition, which also justifies the single-layer transformer experiment:
> > > >
> > > >
> > > > >“Using this decomposition of the data together with RVQs was initially motivated by a series of experimental results. Basing the prediction on the architecture proposed in Gini`es et al. (2025), we trained multiple transformer models to predict high-frequency tokens from their low-frequency counterparts. We conducted nine training runs, varying the model depth from 1 to 15 layers and the input duration from 0.33 s to 5 s. Among all experiments, only shallow models exhibited successful learning, with the single-layer transformer achieving the most reliable training behavior. These results suggest that the task complexity induces weak gradient signals which, when propagated through deeper architectures, lead to systematic collapse.”
> > > >
> > > >
> > > > Second, in response to the reviewer’s remark, we introduced two new experiments that remove semantic partitioning in order to assess its importance:
> > > >
> > > >
> > > > >“To further validate the choice of semantic decomposition and the split-transformer design, we trained two additional models that omit semantic partitioning of the training data: one using three deep transformers identical to those in HP-codecX, and another using a single transformer shared across the three RVQ branches. As shown in Table 4, both variants and the single-layer reference introduced earlier perform markedly worse than HP-codecX, confirming the advantage of incorporating explicit semantic sections.”
> > > >
> > > > ---
> > > >
> > > > Additionally, other edits were made throughout the manuscript to improve clarity and strengthen the justification of our design choices. We invite the reviewer to consult the revised version for the complete set of updates.

---

### Official Review · Reviewer_3hZh · 2025-10-31

**Soundness:** 2
**Presentation:** 2
**Contribution:** 1
**Rating:** 2
**Confidence:** 4

**Summary:**

This paper focuses on the music bandwidth extention (Music BWE), with sampling rate (Fs) = 16kHz as the input, and Fs = 48kHz (full band) as the targeted output.

Technically, this paper adopts the language modeling (LM) approach for audio restoration, which trains an LM with a discrete neural audio codec.

The paper compares the proposed method HP-Codec-X with two open-source BWE model, APOLLO (non-generative model) and AudioSR (a diffusion model trained inside a Mel-VAE latent space, the Mel-to-wav conversion is done by a 48kHz HiFi-GAN vocoder).

However, there are many weaknesses, including missing baseline / ablation study, too limited problem setting, among others.

As a reviewer, I tend to reject this paper.

**Strengths:**

- The model can do 16khz-to-48khz BWE

**Weaknesses:**

## Missing baseline for the codec part
In Sec.3.1.1, authors explicitly mentioned that the architecture is inspired by DAC. Given this relationship, it is essential to evaluate the performance of a standard DAC model in Table.1. Unfortunately, DAC is not included.

As a reader, I would be interested in the benchmark of more standard codecs, including DAC, EnCodec, or more recent MuCodec and SpectroStream. All these models offer pretrained 48kHz or 44.1kHz weights.

Without the comparison with all these prior work, I cannot understand how well the proposed codec is.
## Missing baseline for the LM part
As mentioned above, there have been many open-source 48kHz audio codec.

Before proposing a new LM approach to do BWE, it is essential to train standard LMs on standard codecs such as EnCodec or DAC. Unfortunately, such experiments are not included.

What we can see from the evaluation is that, the proposed method outperforms APOLLO and AudioSR, but we cannot see if the proposed framework is necessary to achieve this performance.

Some speech restoration models trained with standard LM and standard codec:
1. Genhancer: https://www.isca-archive.org/interspeech_2024/yang24h_interspeech.html
2. MaskSR: https://arxiv.org/abs/2406.02092

Another audio language model that can do BWE without finetuning or training
- SpecMaskGIT: https://zzaudio.github.io/SpecMaskGIT/

Obviously, these LM-based speech restoration or audio generation models can be trained to perform BWE by changing the training data, and I would say these models are very standard and contain no tricky parts.

If these simple models already work for music BWE, what is the advantage of HP-Codec-X? The current paper cannot answer this key question.
## Improper problem setting
Music BWE is an important task. However, this paper only handles a fixed 16kHz input, which is too constrained compared to prior arts, and is also too far from real-world scenarios.

For example, AudioSR handles variable bandwidth input.

Considering this point, it is unfair to compare the proposed method (fixed bandwidth input) with other methods that support variable bandwidth input.

**Questions:**

Why standard codecs and standard language model designs were ignored in this paper?

Why only consider Fs=16kHz as  input, instead of variable input bandwidth?

---

> ### Author Response · Authors · 2025-11-20
>
> # Missing Baseline for the Codec Part
>
> We thank the reviewer for highlighting the limited comparison with recent standard codecs. Our work focuses primarily on designing a codec tailored for integration within a bandwidth-extension pipeline, rather than on advancing general-purpose audio compression. For this reason, our initial experiments did not include a broad set of standard codec baselines.
>
> That said, we did verify that our codec achieves competitive reconstruction quality by comparing it to a reference model without semantic sections, itself previously benchmarked against a standard DAC implementation. In the revised version, we will expand this evaluation by including a DAC baseline retrained at 48 kHz and matched to our compression ratio (see Table below). This additional experiment shows that our model performs competitively with DAC and even surpasses it on the ViSQOL metric.
>
> We note that DAC is a widely used codec that has been extensively benchmarked in prior work, which provides indirect but meaningful context for situating our model relative to other standard codecs.
>
> ---
>
> # Missing Baseline for the LM Part
>
> Our objective in this work is to explore whether a VALL-E–style framework (Wang et al., 2023) can be effectively adapted to the bandwidth-extension setting. For this reason, we designed an autoregressive transformer architecture that serves as the basic block of our language model, following the same principles as VALL-E. Introducing a baseline language model fundamentally different from this setup would have required a more complex adaptation process and would not have been well aligned with the structure of our proposed model.
>
> While the alternative approaches suggested by the reviewer may indeed be promising, they rely on a different paradigm and would therefore correspond to a separate research direction. Moreover, speech, music, and environmental audio differ significantly, and models developed for speech (such as those mentioned by the reviewer) do not necessarily transfer well to musical signals, which is the focus of our work. We investigated the possibility of comparing against at least one of the models suggested, but no implementation is currently available. To acknowledge the potential of this alternative direction, we will add a discussion in the revised version of the paper.
>
> ---
>
> # Improper Problem Setting
>
> We acknowledge the reviewer’s concern regarding the fixed 48 kHz sampling rate. This limitation arises from our choice to use full audio codecs as the base architecture, as standard versions of these models are designed for fixed sampling rates and do not trivially support multi-rate operation. A mention of that limitation will be added to the Limitation section in the revised paper.
>
> Nonetheless, extending 16 kHz inputs (with bandwidth up to 8 kHz) to 48 kHz outputs (bandwidth up to 24 kHz) already provides a meaningful and practically useful increase in spectral coverage, supporting the feasibility of our approach. Additionally, the training cost of individual models is relatively modest, making it feasible to train multiple versions at different sampling rates when required.
>
> ---
>
> ## Reconstruction metrics (mean ± standard deviation) for HP-codec
>
> The reference model is a modified version of Ginies et al. (2025) in which the harmonic, percussive, and residual components are removed.
> DAC-48kHz denotes a DAC model retrained on our dataset.
> Both comparison models operate at 48 kHz and use the same compression rate as our model.
>
> | Metric              | HP-codec 16k | HP-codec 48k | Reference 16k | Reference 48k | DAC-48kHz 48k |
> |---------------------|--------------|--------------|----------------|----------------|----------------|
> | **Mel ↓**           | 0.80±0.08    | 0.79±0.05    | 0.70±0.08      | 0.72±0.06      | 0.75±0.08      |
> | **STFT ↓**          | 2.30±0.29    | 2.29±0.29    | 2.11±0.27      | 2.22±0.28      | 2.24±0.28      |
> | **Waveform ↓**      | 0.051±0.015  | 0.052±0.015  | 0.041±0.014    | 0.043±0.014    | 0.041±0.013    |
> | **SI-SDR ↑**        | 6.74±2.53    | 6.30±2.51    | 8.75±2.94      | 8.10±2.91      | 8.79±2.92      |
> | **ViSQOL ↑**        | 4.33±0.09    | 4.33±0.14    | 4.43±0.07      | 4.33±0.17      | 3.92±0.20      |

---

> > ### Author Response · Authors · 2025-11-26
> >
> > We thank the reviewer again for their helpful feedback. We have submitted a revised version of the paper in which all modifications are highlighted in red. Below, we summarize the changes directly addressing the reviewer’s comments.
> >
> > ---
> >
> > ### (1) Additions to Related Work (Section 2.2).
> > In accordance with the reviewer’s suggestions, we now cite and discuss two of the mentioned papers. Specifically, we added the following sentence to Section 2.2:
> >
> >
> > >“A related strategy has recently been explored in speech restoration, where generative language models are trained to predict clean codec tokens from their degraded versions (Li et al., 2024; Yang et al., 2024).”
> >
> >
> > We further added a corresponding discussion in the “Limitations” section:
> >
> >
> > >“An alternative line of research would be to avoid architectural coupling at the codec level and instead draw inspiration from recent approaches such as Li et al. (2024) and Yang et al. (2024). These works employ an off-the-shelf codec to produce incomplete discrete token sequences, and then train a generative language model to recover the corresponding clean representations. Although these techniques have so far been explored only in speech domains, they offer a promising direction for future research on the task considered here.”
> >
> > ---
> >
> > ### (2) Clarification regarding the fixed-sampling-rate constraint (Section 5).
> > We added a paragraph explaining the limitation arising from the use of discrete audio codecs:
> >
> >
> > >“In contrast to various bandwidth-extension systems such as Li & Luo (2025) and Liu et al. (2024a), our model does not support variable input sampling rates and is currently limited to mapping 16 kHz inputs to 48 kHz outputs. This constraint arises primarily from our reliance on discrete audio codecs, which themselves typically operate at fixed sampling rates. Nevertheless, the 16 kHz-48 kHz setting already yields a substantial and practically meaningful improvement in spectral coverage, demonstrating the viability of the proposed framework. Moreover, the relatively low training cost of each model instance makes it feasible to train separate variants for additional sampling rates when needed.”
> >
> > ---
> >
> > ### (3) Expanded evaluation of HP-codec (Section 4.6).
> > We introduced additional paragraphs describing the evaluation setup and motivation for including a retrained DAC baseline:
> >
> >
> > >“To verify that introducing semantic sections does not degrade the reconstruction quality of HP-codec, we evaluate the model on the MUSDB18 (Stöter et al., 2018) test set. For comparison, we adapt the disentangled codec of Giniès et al. (2025) to operate at 48 kHz under identical compression rates, and we retrain a DAC model (Kumar et al., 2023) on 48 kHz audio at the same compression ratio. Since DAC is a widely used baseline with extensive comparisons in the literature, including it provides a clearer sense of how HP-codec aligns with existing methods. The results are summarized in Table 1.”
> >
> >
> > >“These results demonstrate that modifying the RVQ structure to produce a more spectrally informed discrete representation in HP-codec yields performance competitive with both the unmodified reference model and the retrained DAC baseline.”
> >
> > ---
> >
> > Additionally, other edits were made throughout the manuscript to improve clarity and strengthen the justification of our design choices. We invite the reviewer to consult the revised version for the complete set of updates.

---

### Author Response · Authors · 2025-12-02

We thank all reviewers as well as the AC and PC for their time and constructive feedback. We have revised the manuscript accordingly, with all modifications highlighted in red. Below we summarize the main changes introduced in response to the reviewers’ comments:

---

### (1) Expanded evaluation of HP-codec (Section 4.6; Reviewer 3hZh)


To strengthen the empirical validation of HP-codec, we have added a new comparison in Table 1 including a retrained DAC model operating at 48 kHz and matched to the same compression ratio as our model and the reference baseline. These results show that HP-codec achieves competitive reconstruction performance relative to DAC, which itself has been extensively benchmarked in prior work. This provides a clearer indication of how our approach compares against established audio codec baselines.

---

### (2) Clarification of the single-layer transformer experiment and additional validation of semantic decomposition (Section 4.8; Reviewer EHRK)


To better justify the role of semantic decomposition, we introduced two additional ablation settings in Table 4, both preserving the RVQ split but removing semantic decomposition, using either separate transformers for each segment or a single shared transformer for all.
These ablations provide clearer evidence that semantic decomposition is beneficial. We also added a detailed explanation of the training collapse observed in models without semantic decomposition, describing the experimental path that led to this finding.

---

### (3) Additional semantic decomposition analyses (Section 4.8; Reviewer jmoQ)


Following the reviewer’s suggestion, Table 5 now includes expanded decomposition analyses, illustrating the contribution of each semantic section to overall reconstruction quality. These results further clarify the role of individual semantic partitions in the codec’s performance.

---

We also expanded the Limitations section (Section 5) to address concerns raised by Reviewer 3hZh and provided additional motivation and conceptual clarification of our approach in Section 3, addressing questions raised by Reviewer EHRK.


We respectfully invite the AC to consider the paper in light of these substantial revisions and clarifications.

---

### Meta-Review · Area_Chair_LPJx · 2026-01-06

**Summary:**

The main concerns are insufficient baselines (including LM audio restoration baselines and prior bandwidth extension models), validity of the problem setting (fixed sampling rate vs. variable ones used in SoTA models), also the use of lossy codecs for restoration is not adequately justified.

**Reviewer Concerns:**

3hZh:
* missing codec baseline such as standard DAC: partially addressed with a retrained DAC baseline
* missing LM baselines: not addressed, the authors acknowledged but argued it out of scope
* fixed rate vs. SoTA's variable rate: not addressed

EHRK:
* single layer transformer used in baseline vs. deeper models for the proposed model: the authors argued deeper model collapsed, which is a concerning.
* missing comparisons to Grumiaux & Lagrange (2023) which this work borrow's part for data: not addressed due to failed to install the package, which is not convincing

jmoQ:
* why use lossy codec for bandwidth extension: the authors' justification is not very convincing
* requested more ablations on each branches' contribution: the authors added additional results
* algorithm detail clarification: the authors added more details

**Reviewer Scores:**

3hZH: 2, no change
EHRK: 4, no change
jmoQ: 4, no change

---

### Decision · Program_Chairs · 2026-01-26

Reject